# Path-Specific Counterfactual Fairness via Dividend Correction

**Daisuke Hatano**                                                    *daitoku.hatano@gmail.com*
*RIKEN Center for Advanced Intelligence Project*

**Satoshi Hara**                                                          *satohara@uec.ac.jp*
*The University of Electro-Communication*

**Hiromi Arai**                                                          *hiromi.arai@riken.jp*
*RIKEN Center for Advanced Intelligence Project*

**Reviewed on OpenReview:** *https://openreview.net/forum?id=RXoSmiyObR*

## Abstract

Counterfactual fairness is a fundamental principle in machine learning that allows the analysis of the effects of sensitive attributes in each individual decision by integrating the knowledge of causal graphs. An issue in dealing with counterfactual fairness is that unfair causal effects are often context-specific, influenced by religious, cultural, and national differences, making it difficult to create a universally applicable model. This leads to the challenge of dealing with frequent adaptation to changes in fairness assessments when localizing a model. Thus, applicability across a variety of models and efficiency becomes necessary to meet this challenge. We propose the first efficient post-process approach to achieve path-specific counterfactual fairness by adjusting a model's outputs based on a given causal graph. This approach is model-agnostic, prioritizing on flexibility and generalizability to deliver robust results across various domains and model architectures. By means of the mathematical tools from cooperative game theory, the Möbius inversion formula and dividends, we demonstrate that our post-process approach can be executed efficiently. We empirically show that our proposed algorithm outperforms existing in-process approaches for path-specific counterfactual fairness and a post-process approach for counterfactual fairness.

## 1 Introduction

The rise of machine learning has significantly transformed decision making processes in daily life. However, in high-stakes applications such as healthcare (Ahsan et al., 2022), criminal justice (Brennan et al., 2009), hiring (Hoffman et al., 2018), and lending (Khandani et al., 2010), fairness becomes a critical consideration. As these domains directly impact individuals, it is essential to ensure that decisions made by machine learning models are both fair and unbiased. Recently, the machine learning community has developed various statistical criteria to formalize fairness notions (Dwork et al., 2012; Feldman et al., 2015; Hardt et al., 2016). However, most criteria are enforced by introducing fairness constraints during the model training process, often creating a trade-off between fairness and accuracy.

To relax this trade-off, Kusner et al. (2017) introduced *counterfactual fairness*, which aims to ensure fair decisions without sacrificing accuracy too much by focusing on the causal relationship between a sensitive attribute and an outcome. Their framework assumes that any causal effect of the sensitive attribute on the outcome as inherently problematic and aims to eliminate any influence that the sensitive attribute might have on the outcome. However, failing to distinguish between fair and unfair pathways can lead to the removal of legitimate effects. This oversimplification risks undermining the model's fairness. In cases involving physical strength, outcomes such as suitability for physically demanding jobs may depend on

sensitive attributes due to biological differences. Removing the causal path from gender to physical strength could result in unfairness. This complexity highlights the limitation of conventional fairness notions, which struggle to distinguish between fair and unfair causal pathways. The fairness in such scenarios cannot be adequately captured using traditional statistical metrics like demographic parity in group fairness (Dwork et al., 2012; Verma & Rubin, 2018) or even counterfactual fairness. To address this issue, Chiappa (2019); Wu et al. (2019b) introduced *path-specific counterfactual fairness*, which aims to eliminate only the unfair causal effects of the sensitive attribute while preserving the fair ones. This framework has inspired a range of fairness algorithms designed to achieve path-specific counterfactual fairness (Zhang et al., 2017; Nabi & Shpitser, 2018; Chiappa, 2019; Wu et al., 2019b; Chikahara et al., 2022).

In this study, we propose a *post-process* method to achieve path-specific counterfactual fairness. A significant challenge in achieving path-specific counterfactual fairness is that unfair pathways can be influenced by specific religions, cultures, or countries, making it difficult to develop a universally applicable model. For instance, fairness assessments often depend on regional ethical standards, such as General Data Protection Regulation (GDPR) [1] in Europe, with different standards prevailing in other regions. This variability requires adaptations of fairness assessments to local contexts, highlighting the need for an efficient and flexible fairness method. In contrast, most counterfactual fairness methods rely on in-process approaches, which require building fairness-aware models from scratch. To overcome this limitation, we propose a method that achieves counterfactual fairness by post-processing the outputs of existing models, enhancing the adaptability of fairness assessments across diverse cultural and religious settings.

In our method, we leverage the set function framework, as the predictive model's output can be interpreted as a set function when the sensitive attribute is binary. As conventional set functions cannot accurately capture the causal effect of the sensitive attribute on the output, it is essential to implement a framework that can isolate causal effects from correlations. By introducing an *interior operator*, we construct a causally-aware set function that effectively captures causal dependencies. We then clarify the relationship between the causally-aware set function and path-specific counterfactual fairness. To deconstruct the causally-aware set function into meaningful parts, we introduce a specialized *Möbius inversion formula* (Stanley, 2011; Algaba et al., 2004) that accounts for hierarchical constraints that reflect causal relationships. This enables us to decompose it into the sum of *dividends* (Harsanyi, 1958). The dividend can be intuitively understood as additional value created by modifying specific attributes, after considering the overlapping contributions of their smaller attribute subsets. We elucidate each dividend corresponds to a well-known causal effect and achieve fairness by removing dividends associated with unfair pathways.

One of our contributions is to propose the first post-process approach designed for path-specific counterfactual fairness. To this end, we introduce a novel causally aware set function by leveraging concepts from combinatorics to handle precedence structures. Through the decomposition of this set function, we elucidate the relationship between its components and various causal effects, corresponding to total direct effect, path-specific effect, and mediated interaction. This decomposition enables a more precise analysis of how individual causal pathways contribute to the overall effects. Another contribution is that the execution time of our algorithm is linear with respect to the size of the given causal graph, as it is based on running the model inference a number of times proportional to the number of nodes in the graph. Additionally the algorithm requires model training time for each mediator. Despite this additional computation, our algorithm remains efficient compared to the existing in-process approaches, as shown in the experimental evaluation.

## 2 Related work

Approaches for ensuring fairness in machine learning models typically fall into three main categories. (i) Pre-processing approach entails modifying the training data to remove biases before the model training stage, e.g., by group-wise data rescaling (Feldman et al., 2015), data re-labeling (Kamiran & Calders, 2012), and instance re-weighting (Kamiran & Calders, 2012). This approach is model-agnostic because it focuses on the input data rather than the structure of the model. However, pre-processing alone may not fully satisfy the fairness condition in the final model prediction. (ii) In-processing approach adapts the learning algorithm to prevent bias, e.g., by incorporating regularization terms to penalize unfair outcomes (Kamishima et al.,

---

[1]https://eur-lex.europa.eu/eli/reg/2016/679/oj

2012; Fukuchi & Sakuma, 2014) or by introducing fairness constraints (Zafar et al., 2017; Donini et al., 2018; Agarwal et al., 2018). In-process approaches tend to be more complex and computationally expensive, but can directly control biases in the learning process. They are often limited to models that support gradient-based optimization, making them difficult to apply to certain models, such as decision trees. (iii) Post-processing approach adjusts the predictions of the trained model to ensure fairness by modifying the outputs (Kamiran et al., 2012; Hardt et al., 2016). This approach is computationally efficient and flexible, as it modifies the output of a given model and can be applied to any learning model after training. However, post-processing approaches often compromise accuracy more than in-process approaches.

Fairness studies within the context of causality have mainly concentrated on in-process approaches. Most fairness criteria related to causality, such as counterfactual and path-specific counterfactual fairness, have been developed within this framework. Some approaches try to ensure the fairness by integrating fairness criteria into the model's optimization process by adding fairness constraints during training (Kusner et al., 2017; Nabi & Shpitser, 2018; Wu et al., 2019b; Chikahara et al., 2021). Another approach focuses on modifying individual data points to ensure fair outcomes (Chiappa, 2019). Applying in-process approaches in settings where unfair pathways require frequent modification presents several limitations: (i) Adding fairness constraints during training makes the model optimization process more complex, which increases the time required to train models. (ii) Fairness constraints restrict a model's flexibility, especially when certain learning models cannot easily accommodate these constraints within their optimization framework, such as decision trees, k-NN, clustering algorithms, and ensemble models that lack gradient-based optimization frameworks. Post-process approaches can overcome these limitations and do not alter the model training phase, allowing for flexibility and computational efficiency. The only post-process approach was proposed by Wu et al. (2019a) to address counterfactual effects, but it is limited to counterfactual fairness, and the computational complexity of their LP-based approach increases exponentially with the size of a causal graph. In contrast, our research presents a novel approach for achieving path-specific counterfactual fairness that runs in linear time relative to the size of the causal graph.

Our main mathematical tools, the Möbius inversion formula and dividend, are widely used in the field of cooperative game (Algaba et al., 2004; Ieong & Shoham, 2005). These notions are closely related to the Shapley value (Shapley, 1953), which is now widely used in machine learning, e.g., for feature selection (Cohen et al., 2005; Sun et al., 2012), data valuation (Jia et al., 2019; Ghorbani & Zou, 2019), and explainability (Lundberg & Lee, 2017; Sundararajan & Najmi, 2020) (see (Rozemberczki et al., 2022)). Although these mathematical tools are from the same field, the way we use the Möbius inversion formula and dividend are different from the aforementioned studies. While we use the Möbius inversion formula and dividend for post-hoc model correction, Shapley value is used to quantify the contribution of each individual (such as feature or data instance) in the aforementioned studies. Our finding will provide a new possible connection between machine learning and cooperative game theory beyond the Shapley value.

Feasible sets are a notion introduced in Antimatroids (Korte et al., 2012), and it also has been studied in conjunction with the Möbius inversion formula (van den Brink, 2017; Algaba et al., 2004). Antimatroid is used as a hierarchical structure among players called permission structure (van den Brink & Gilles, 1996; van den Brink, 1997), and the Möbius inversion formula is used as a tool to analyze the properties of the Shapley value. To our knowledge, we will be the first to employ antimatroids and the Möbius inversion formula in the analysis of causality.

## 3 Preliminary

### 3.1 Path-specific counterfactual fairness

We introduce key notations to formalize concepts in causality and path-specific counterfactual fairness. Causality is often expressed using a directed acyclic graph, known as a *causal graph*. Let $G$ be a directed acyclic graph, where a set of nodes corresponds to features $X$, and a set of directed edges indicates causal relationships between them. For each direct edge $i \rightarrow j$ where $i, j \in X$, $i$ is referred to as a *parent* of $j$ while $j$ is referred to as a *child* of $i$. The set of nodes that can reach node $i$ is referred to as the ancestors of $i$.

Figure 1 displays an example of a causal graph illustrating hiring decisions for physically demanding positions. In this graph, the endogenous variables $A, Q, D, M \in X$ represent gender, qualification, faculty type, and physical strength, respectively. The qualification represents the skills and knowledge, the faculty type refers to the specific academic department, such as STEM (Science, Technology, Engineering, and Mathematics) fields, and the physical strength describes an individual's physical ability or capacity. Gender $A$ is a *sensitive attribute*, while the features $D$ and $M$ are referred to as mediators, which are influenced by $A$ through causal pathways. Nodes $e_D$ and $e_M$ indicate exogenous variables that work as error variables. For simplicity, we do not show exogenous variables for each mediator in the following causal graphs. In the path-specific counterfactual setting, not all influences from the sensitive attribute are necessarily unfair. In Figure 1, the causal path through $A \to M \to Y$ is considered fair, as it reflects a relationship influenced by innate factors between the gender $A$ and the physical strength $M$. Conversely, the pathways $A \to Y$ and $A \to D \to Y$ are considered unfair, as they involve social or cultural factors that may lead to discriminatory outcomes. For instance, traditionally, men tend to enroll in engineering, physics, and computer science programs, while women are more inclined to choose fields such as nursing, education, and social work. Note that every edge in the causal graph belongs to either fair or unfair pathways. While the fairness of pathways may frequently change depending not only on the location where the predictive model is applied, but also on the specific context or situation, the unfair pathways in this setting are fixed.

We concentrate on *path-specific counterfactual fairness* as outlined by (Chiappa, 2019; Wu et al., 2019a), which seeks to isolate the effect of unfair pathways. The set of unfair pathways is denoted as $\pi$, i.e., $\pi = \{A \to Y, A \to D \to Y\}$ in the example. To establish the definition of path-specific counterfactual fairness, we will examine path-specific causal effects as described by (Avin et al., 2005; Albert & Nelson, 2011) that measures the extent to which an attribute influences other features through causal paths. The path-specific causal effect can be obtained by comparing two predictions generated by modifying inputs $X$. The focus is on a situation in which an individual accepts the aforementioned fairness while being assessed for hiring based on the prediction of a machine learning model. Let $f_D$ and $f_M$ be causal models of the mediators $D$ and $M$ from which the

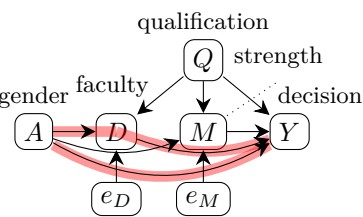

Figure 1: An example of a causal graph indicating hiring decisions for physically demanding jobs (Chikahara et al., 2022). The two red paths denote unfair pathways.

values of $D$ and $M$ can be estimated, i.e., $D = f_D(A, Q, e_D)$, and $M = f_M(A, Q, e_M)$. In addition, the value of $Y$ can be obtained using the predictive model $f$, given by $Y = f(A, Q, D, M)$. Assuming that the value of $A$ for a given data point $X$ is 0, the path-specific counterfactual effect (PSE) can be defined as the difference between two predictions $Y_{A \Leftarrow 1 \| \pi} - Y_{A \Leftarrow 0}$, when the value of the sensitive attribute $A$ flips from 0 to 1 under unfair pathways $\pi$, where $Y_{A \Leftarrow 1 \| \pi}$ and $Y_{A \Leftarrow 0}$ are called potential outcomes, expressed as follows:

$$Y_{A \Leftarrow 1 \| \pi} = f(1, Q, f_D(1, Q, e_D), f_M(0, Q, e_M)), \quad Y_{A \Leftarrow 0} = f(0, Q, f_D(0, Q, e_D), f_M(0, Q, e_M)),$$

where $Y_{A \Leftarrow 1 \| \pi}$ represents a prediction when modification of $A$ only affects unfair pathways, implying that the value of $M$ remain unaffected by the change. Specifically, this means $M = f_M(0, Q, e_M)$. On the contrary, as $Y_{A \Leftarrow 0}$ represents the prediction for the original data point $X$, it is not influenced by $\pi$. Employing these notations, path-specific counterfactual fairness can be defined using PSE, given as follows:

**Definition 3.1** (Path-specific counterfactual fairness (PSCF) (Chiappa, 2019; Wu et al., 2019b)). Given the unfair pathways $\pi$ of a causal graph $G$ and causal models $f_D$ and $f_M$, a classifier $f$ achieves path-specific counterfactual fairness if for any value $x$ of $X$, it holds that

$$\mathbb{E}_{Y_{A \Leftarrow 1 \| \pi}, Y_{A \Leftarrow 0}}[Y_{A \Leftarrow 1 \| \pi} - Y_{A \Leftarrow 0} \mid X = x] = 0. \tag{1}$$

Since the PSCF is designed for binary sensitive attributes, it needs to be generalized to accommodate non-binary sensitive attributes. For simplicity, we assume in the following section that the value of the sensitive attribute is flipped from 0 to 1 without loss of generality.

## 3.2 A decomposition of the total effect into path-specific effects

Using the notations of Figure 1, we introduce some causal effects and some of its decomposition. The total effect of alternation on the sensitive attribute $A$ is defined as

$$\text{TE} := Y_{A \leftarrow 1} - Y_{A \leftarrow 0} = f(1, Q, f_D(1, Q, e_D), f_M(1, Q, e_M)) - f(0, Q, f_D(0, Q, e_D), f_M(0, Q, e_M)).$$

For simplicity, we will write $f(A, Q, f_D(A, Q, e_D), f_M(A, Q, e_M))$ as $Y_{A D_A M_A}$, since $Q$, $e_D$, and $e_M$ are not affected by the change in $A$. TE can be decomposed as $\text{TE} = \{Y_{1 D_1 M_1} - Y_{0 D_1 M_1}\} + \{Y_{0 D_1 M_1} - Y_{0 D_0 M_0}\}$, where the first term is the *total direct effect*, denoted by TDE, and the second term is the *pure indirect effect*, denoted by PIE (Robins & Greenland, 1992; VanderWeele, 2013). TDE is the effect of flipping sensitive attribute $A$ from 0 to 1 while considering the effect of changes in $A$ on the mediators. This allows us to capture the interaction effect between $A$ and the mediators $D$ and $M$. In contrast, PIE is the effect of changing the values of the mediators while keeping $A$ to 0. This means PIE represents the indirect effect, excluding the interaction effect between $A$ and the mediators $D$ and $M$.

PIE can be further decomposed into path-specific effects. However, Avin et al. (2005) showed that the sum of path-specific effects is not equivalent to the total effect, as certain components are missing. Taguri et al. (2018) later proved the missing components are due to mediated interaction. By introducing this effect, PIE can be decomposed into the sum of path-specific effects and mediated interaction, given by:

$$\text{PIE} := \{Y_{0 D_1 M_1} - Y_{0 D_0 M_1}\} + \{Y_{0 D_1 M_1} - Y_{0 D_1 M_0}\} - \{Y_{0 D_1 M_1} - Y_{0 D_1 M_0} - Y_{0 D_0 M_1} + Y_{0 D_0 M_0}\}, \quad (2)$$

where the fist term is the path-specific effect via the pathway $A \rightarrow D \rightarrow Y$, the second term is the *path-specific effect* via the pathway $A \rightarrow M \rightarrow Y$, and the third term is the *mediated interaction* between $A \rightarrow D$ and $A \rightarrow M$. The path-specific effect measures the effect of an intervention on an outcome along a specific causal path involving mediators. For a pathway of $A \rightarrow D \rightarrow Y$ and a target mediator $D$, the path-specific effect can be calculated as the difference between the predicted values when $M$ takes on a specific value influenced by the the exposure and the predicted value when $M$ is set to its original data point. Mathematically, the path-specific effects via mediator $D$ and $M$ are defined by

$$\text{PSE}_D := Y_{0 D_1 M_1} - Y_{0 D_0 M_1}, \ \text{PSE}_M := Y_{0 D_1 M_1} - Y_{0 D_1 M_0},$$

respectively[2]. The last term of equation 2 is referred to as the *mediated interaction*, given by

$$\text{MI}_{DM} := -(Y_{0 D_1 M_1} - Y_{0 D_1 M_0} - Y_{0 D_0 M_1} + Y_{0 D_0 M_0}).$$

Then PIE can be represented as $\text{PIE} = \text{PSE}_D + \text{PSE}_M + \text{MI}_{DM}$, and we have

$$\text{TE} = \text{TDE} + \text{PSE}_D + \text{PSE}_M + \text{MI}_{DM}.$$

## 4 Problem setting and motivating scenario

The motivating scenario is the release of an AI-based decision-making system that involves two main stakeholders: an AI model publisher and an auditor. The publisher aims to deploy an AI-decision making service built on their predictive model $f$. The auditor's role is to assess this model to determine whether it meets fairness criteria. To ensure the integrity and transparency of the service, the publisher must publicly disclose the causal graph $G$ that underpins the predictive model $f$. The auditor will evaluate the fairness of $f$ based on this pre-registered causal graph $G$, regardless of its accuracy. In addition, the auditor should learn the causal models of mediators, such as $f_D$ and $f_M$, to accurately estimate their values when altering the sensitive attribute. This is important because, when the publisher can define their causal models, the values of mediators can be adjusted as desired when altering the sensitive attribute.

For instance, if the publisher sets up a model where the sensitive attribute (gender) influences the mediator (income), they can adjust how much income changes when the sensitive attribute flips from male to female.

---

[2]PSE is alternatively defined as $PSE_D := Y_{1 D_1 M_0} - Y_{1 D_0 M_0}$ or $Y_{1 D_1 M_1} - Y_{1 D_0 M_1}$ in (Taguri et al., 2018; Huang & Cai, 2016; Huang & Yang, 2017; Steen et al., 2017)

This adjustment can lead to a manipulated fairness outcome, as the publisher has the ability to tweak the system to make it appear fairer or more biased based on how they model the interaction between gender and income. This ability to influence the intermediate mediators could distort the auditor's ability to evaluate fairness accurately.

Therefore, in this paper, we define the following problem setup: (i) A causal graph $G$, unfair pathways $\pi$, and a predictive model $f$ are given, and (ii) Causal models of mediators, e.g., $f_D$ and $f_M$, need to be prepared by the auditor. With these settings, our objective is as follows.

**Objective.** Our aim is to develop a *post-process approach* that modifies the output $f(x)$ of a given model $f$ for every individual data $x$, such that equation 1 holds.

## 5 Proposed algorithm

We first briefly describe the idea of our algorithm.

The key components of our algorithm is feasible sets, dividends, and Möbius inversion formula. The feasible sets represent subsets that capture causal relationships, and the dividends quantify additional contribution by modifying specific attributes, accounting for the overlapping contributions of their smaller attribute subsets. The Möbius inversion formula serves as the method for computing these dividends. The proposed algorithm operates through five steps, as shown in Figure 2. (i) A prediction $f(x)$ of a data point $x$ is translated into a set function $v_{\mathcal{F}}$ that incorporates causal relationship, as we focus on the binary state of a sensitive attribute. To achieve this, we employ *feasible sets* to represent causal relationships and an *interior operator* to incorporate the feasible sets into the set function. (ii) The value of $v_{\mathcal{F}}$ is decomposed into the causal effects of individual causal paths through the *Möbius inversion formula*, allowing us to decompose the value into the sum of dividends. We elucidate that the dividends of any subset in $v_{\mathcal{F}}$ correspond to TDE, PIE, or MI, as described in Section 3.2. (iii) Using this relationship, fair dividends $\Delta_{v'_{\mathcal{F}}}$ are obtained by nullifying dividends $\Delta_{v_{\mathcal{F}}}$ of every unfair pathway. (iv) A fair set function $v'_{\mathcal{F}}$ can be constructed from the fair dividends $\Delta_{v'_{\mathcal{F}}}$, exploiting the property of the Möbius inversion formula, which ensures any set function has a unique representation in terms of dividends. (v) Ensure that the value $v'_{\mathcal{F}}$ obtained in Step (iv) corresponds to the desired value $f'(x)$, given that the translation from a set function to a prediction on $x$ can be performed without loss.

### 5.1 Construction of causally aware set functions

The aim of this section is to create a *causally aware set function* $v_{\mathcal{F}}$, using a given predictive model $f$ and a particular data point $x$, which is related to Steps (i) and (v) in Figure 2. We construct $v_{\mathcal{F}}$ that satisfies the following conditions: temporal precedence and unconfoundedness. These conditions are necessary for inferring causal effect, as described in Pearl (2009). Temporal precedence ensures that the cause occurs before the effect, which is required for establishing

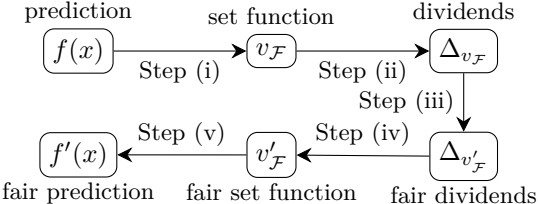

Figure 2: The flow of our algorithm.

a causal relationship. Unconfoundedness ensures that there are no unmeasured confounders that influence both the cause and the effect. In other words, it means that any confounders must either be measured or accounted for. Unfortunately, conventional set functions do not hold the above properties. In the following, we describe how to construct a causally aware set function $v_{\mathcal{F}}$ to satisfy the desired properties.

We begin by creating a conventional set function $v_{x,r} : 2^N \to \mathbb{R}$ of $f(x)$, where $N$ stands for a set of $n$ features in $X$ excluding the target variable $Y$ and $x$ and $r$ indicate feature values of a given data point and a point of reference, respectively. For a given subset $S \subseteq N$, the value $v_{x,r}(S)$ can be represented by analyzing the impact of the absence of features in the complement set $N \setminus S$, given by $v_{x,r}(S) := f(r_S, x_{S^c})$ for any $S \subseteq N$, where $r$ is an $n$-dimensional vector called a *reference value*, and $(r_S, x_{S^c})$ indicates an $n$-dimensional vector

whose $i$-th component is $r_i$ if $i \in S$ and $x_i$ otherwise[3]. $S^c$ represents the complement of $S$, i.e., $S^c = N \setminus S$. The value $v_{x,r}(N \setminus i)$ indicates the prediction obtained by changing the value of $i$ from $x_i$ to $r_i$. In what follows, unless otherwise specified, we will abbreviate $v_{x,r}$ as $v$.

The set function $v$ constructed above is not compatible in causal contexts because it does not accurately present causal relationships. To implement the relationship into the set function, we employ an *interior operator* (Korte et al., 2012), given by $\mathrm{int}_{\mathcal{F}} : 2^N \to \mathcal{F}$, where $\mathcal{F}$ is a family of subsets of $N$ referred to as *feasible sets*, which is often referred to as ancestral sets in the causal context. Every feasible set $S \in \mathcal{F}$ satisfies the requirement that if a node is included in $S$, then all its parent nodes are also into $S$. The interior operator is formally defined by

$$\mathrm{int}_{\mathcal{F}}(S) := \bigcup \{T \subseteq S \mid T \in \mathcal{F}\}, \quad \forall S \subseteq N. \tag{3}$$

With these notations, a causally aware set function $v_{\mathcal{F}}$ is defined as follows:

$$v_{\mathcal{F}}(S) := v(\mathrm{int}_{\mathcal{F}}(S)), \quad \forall S \subseteq N. \tag{4}$$

Intuitively, the interior operator modifies an infeasible condition, where some children of absent features are present, into a feasible condition, where no children of absent features are present. Hence, the causally aware set function $v_{\mathcal{F}}$ has a capacity to identify causal relationships.

**Example 1.** *Feasible sets are given by $\mathcal{F} = \{\emptyset, \{A\}, \{Q\}, \{A, Q\}, \{A, Q, D\}, \{A, Q, M\}, \{A, Q, D, M\}\}$ in Figure 1. In Figure 3 (b), given $\{Q, D, M\}$, the interior operator returns $\{Q\}$, and the value of $v_{\mathcal{F}}(\{Q, D, M\})$ corresponds to $v(\{Q\}) = f(x_A, r_Q, x_D, x_M)$.*

Figure 3 (a) demonstrates the operation of a set function $v$ and an interior operator when $N \setminus \{A\} = \{Q, D, M\}$ is given. Evaluating the value of $\{Q, D, M\}$ on $v$ does not reflect causality as nodes $D$ and $M$ are children of the absent feature $A$. This implies that the values of the nodes $D$ and $M$ remain unchanged unless the parent's value $A$ changes. Alternatively, the interior operator provides a feasible set $\{Q\}$, which exhibits a causal relationship.

The reference value $r$ plays an important role in the causally aware set function $v_{\mathcal{F}}$ to measure the causal effect size. The reference value accounts for how changes in the sensitive attribute influence the values of other features, ensuring a proper estimation of causal effects. Since the true causal model is not known, we need to learn the mediators' models $f_D, f_M$ to accurately estimate the reference values. The causal models $f_D$ and $f_M$ are learned through the following process. First, we estimate the value of the exogenous variables for each data point $x$. Then we use these estimates to learn causal models for the mediators. For estimating the exogenous variables, we can apply several estimation algorithms, such as Maximum Likelihood Estimation (MLE) via the Expectation-Maximization (EM) algorithm and Bayesian inference using methods like Gaussian approximation or the Markov chain Monte-Carlo (MCMC) method. The detailed approach used in our experiments is written in Section C.3. After estimating the value of exogenous variables, causal models for mediators can be learned using appropriate learning models, such as linear regression or neural networks. As a result, the reference values for the mediators can be obtained from the learned causal models and the estimated exogenous variables. For variables that are not affected by the sensitive attribute, their reference values remain unchanged from the given data point $x$.

**Example 2.** *In Figure 1, by changing $A$ from 0 to 1, the values of $D$ and $M$ are also affected. They determine their reference value by examining their causal models $f_D$ and $f_M$, which are obtained by $r_A = 1$, $r_D = f_D(r_A, x_Q)$, and $r_M = f_M(r_A, x_Q)$. On the contrary, as the value of $Q$ is not affected by the value of $A$, and its reference value remains unchanged, i.e., $r_Q = x_Q$.*

The causally aware set function $v_{\mathcal{F}}$ satisfies the two conditions of temporal precedence and unconfoundedness. Temporal precedence implies that a cause occurs before its effect. We can easily check that $v_{\mathcal{F}}$ supports the temporal precedence from Figure 3, since the existence of child nodes, e.g., $D$ and $M$, depends on the existence of their parent nodes $A$ and $Q$. In other words, changes in $A$ or $Q$ lead to changes in their children,

---

[3]Although the set function is defined as $v_{x,r}(S) := f(x_S, r_{S^c})$ in (Lundberg & Lee, 2017), in this paper, we swap the positions of $x$ and $r$ for convenience.

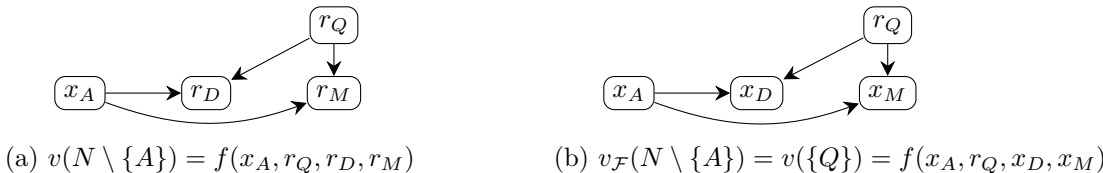

(a) $v(N \setminus \{A\}) = f(x_A, r_Q, r_D, r_M)$        (b) $v_{\mathcal{F}}(N \setminus \{A\}) = v(\{Q\}) = f(x_A, r_Q, x_D, x_M)$

Figure 3: How the conventional set function $v$ and the causally aware set function $v_{\mathcal{F}}$ work.

$D$ and $M$, indicating a causal direction. Unconfoundedness means that the non-existence of unmeasured confounders. In our problem setting, the reference values $r$ of any confounder are equivalent to its data point $x$. This implies that for any $S \setminus N$ and any confounder $i \in N \setminus S$, $v_{\mathcal{F}}(S) = v_{\mathcal{F}}(S \cup i)$. This aligns with ignorability by satisfying that confounders outside of $S$ have no impact on the causal effect.

In this section we have focused on changing the gender attribute from 0 to 1. However, this approach is not limited to binary attributes; it also applies to sensitive attributes that can take more than two values from a discrete set. Nonetheless, when the sensitive attribute has more than two values, it is important to define the fairness criteria carefully, as they may not be uniquely determined in such cases.

## 5.2 Decomposition into the sum of causal effects

The purpose of this section is to clarify the connection between $v_{\mathcal{F}}$ and causal effects introduced in Section 3.2. The connection allows us to ensure the path-specific counterfactual fairness by removing the corresponding causal effects of unfair pathways, which falls under Steps (ii), (iii), and (iv) in Figure 2. We first show that $v_{\mathcal{F}}$ can be presented through the sum of causal effects by using the *Möbius inversion formula* (Stanley, 2011), to decompose an obtained set function $v$ into the sum of dividends, which are a particular form of synergy effect. The Möbius inversion formula of a set function structured by feasible sets $v_{\mathcal{F}}$ was proposed by Algaba et al. (2004), given by

$$v_{\mathcal{F}}(S) = \sum_{T \subseteq S} \Delta_{v_{\mathcal{F}}}(T), \quad \forall S \subseteq N, \tag{5}$$

where $\Delta_{v_{\mathcal{F}}} : 2^N \to \mathbb{R}$ is called Möbius transformer of $v_{\mathcal{F}}$ or is often referred to as *(Harsanyi) dividend* in the cooperative game context (Harsanyi, 1958). For each $T \subseteq N$, the dividend $\Delta_{v_{\mathcal{F}}}(T)$ is computed by

$$\Delta_{v_{\mathcal{F}}}(T) = \begin{cases} \sum_{T \setminus en(T) \subseteq U \subseteq T} (-1)^{t-u} v_{\mathcal{F}}(U), & \text{if } T \in \mathcal{F}, \\ 0, & \text{otherwise,} \end{cases} \tag{6}$$

where $t := |T|$, $u := |U|$, and $en(T)$ is a set of nodes that does not have any child in $T$. Intuitively, $\Delta_{v_{\mathcal{F}}}(T)$ indicates the additional accuracy obtained by modifying attributes $T$, after accounting for the overlapping accuracy contributed by the subsets of $T$.

We then explore the relationship between path-specific counterfactual fairness and dividends. The lemma suggests that the PSE can be computed leveraging the dividends representation.

**Lemma 5.1.** *Let $\mathcal{F}_\pi$ be a feasible set of a DAG induced by unfair pathways $\pi$. Then the prediction $v(N)$ is path-specific counterfactual fair if and only if its corresponding causally-aware set function $v_{\mathcal{F}_\pi}$ holds that*

$$\sum_{S \in \mathcal{F}_\pi \setminus \{\emptyset\}} \Delta_{v_{\mathcal{F}_\pi}}(S) = 0. \tag{7}$$

*Proof sketch.* To connect the dividend representation and the definition of PSCF, given in equation 1, we define $r'$ as the reference value affected only through unfair pathways and $\mathcal{F}_\pi$ as feasible sets of a DAG induced by unfair pathways, as exemplified in Example 6. Let $v_{x,r',\mathcal{F}}$ denote a set function obtained by replacing $r$ into $r'$ from $v_{\mathcal{F}}$, and $v_{x,r,\mathcal{F}_\pi}$ denote a set function obtained by replacing $\mathcal{F}$ into $\mathcal{F}_\pi$ from $v_{\mathcal{F}}$. We show that $\sum_{S \in \mathcal{F}} \Delta_{v_{x,r',\mathcal{F}}}(S) = \sum_{S \in \mathcal{F}} \Delta_{v_{x,r,\mathcal{F}_\pi}}(S)$ using the binomial theorem. The statement is obtained by using this relationship. $\square$

Lemma 5.1 indicates that we can obtain a fair prediction by computing the dividends of all feasible sets on unfair pathways $\mathcal{F}_\pi$ and setting the value of the dividends to zero. Thus, we obtain the next lemma.

**Lemma 5.2.** *Given unfair pathways $\pi$, a modified prediction $f'(x)$ achieves path-specific counterfactual fairness if it holds that*

$$f'(x) = f(x) + \sum_{S \in \mathcal{F}_\pi \setminus \{\emptyset\}} \Delta_{v_{\mathcal{F}_\pi}}(S). \tag{8}$$

*Proof.* We first set the sum of dividends of $f'(x)$ as that of $f(x)$, that is, $\sum_{S \in \mathcal{F}} \Delta_{v_{\mathcal{F}}}(S)$. By setting the values of the dividends involving unfair pathways $\mathcal{F}_\pi$ to zero, we satisfy the condition of Lemma 5.1, i.e., the value of $f(x)$ matches $f'(x) - \sum_{S \in \mathcal{F}_\pi \setminus \{\emptyset\}} \Delta_{v_{\mathcal{F}_\pi}}(S)$. Hence, we achieve the desired result. $\square$

We can interpret each dividend as the notation of causal effects introduced in Section 3.2. The next proposition outlines this interpretation.

**Proposition 5.3.** *The dividend $-\Delta_{v_{\mathcal{F}}}(S)$ shows the total direct effect if $S$ is a root node of $G$, the path-specific effect if $S$ indicates single pathway, and the mediated interaction if $S$ is the union of multiple pathways.*

*Proof sketch.* We can easily prove the proposition by correlating each equation of causal effects and the calculations of corresponding dividends. Here, we demonstrate the causal effects of some specific subsets, using Example 1. The total direct effects in altering the value of $A$ is given by $-\Delta_{v_{\mathcal{F}}}(\{A\}) = v_{\mathcal{F}}(\emptyset) - v_{\mathcal{F}}(\{A\}) = Y_{1D_1M_1} - Y_{0D_1M_1} = \text{TDE}$. When a feasible set is singleton, i.e., $S \in \mathcal{F}$ with $|S| = 1$, the corresponding dividend corresponds to TDE. The path-specific effect through $D$ is obtained by $-\Delta_{v_{\mathcal{F}}}(\{A, D\}) = v_{\mathcal{F}}(\{A\}) - v_{\mathcal{F}}(\{A, D\}) = Y_{0D_1M_1} - Y_{0D_0M_1} = \text{PSE}_D$. When a feasible set is constructed by a single pathway, e.g., $S = \{A, D\}, \{A, M\}$, the corresponding dividend is equivalent to PSE. The mediated interaction between $M$ and $D$ is calculated by $-\Delta_{v_{\mathcal{F}}}(\{A, D, M\}) = -(v_{\mathcal{F}}(\{A\}) - v_{\mathcal{F}}(\{A, D\}) - v_{\mathcal{F}}(\{A, M\}) + v_{\mathcal{F}}(\{A, D, M\})) = -(Y_{0D_1M_1} - Y_{0D_1M_0} - Y_{0D_0M_1} + Y_{0D_0M_0}) = \text{MI}_{DM}$. When a feasible set is constructed by the union of multiple pathways, e.g., $S = \{A, D, M\}$, the corresponding dividend is equivalent to MI. The details are provided in the appendix. $\square$

With Lemma 5.1 and Proposition 5.3, a modified prediction $f'(x)$ is obtained by computing the dividends of all feasible sets on unfair pathways $\mathcal{F}_\pi$. Intuitively, these statements indicate that the prediction $f(x)$ is path-specific counterfactual fair if causal effects from the sensitive attribute are ineffective to the prediction. Furthermore, we note that our approach can independently quantify the influence of unfair pathways by introducing specific parameters to represent their respective degrees of impact. By doing so, we enable to adjust for the varying impact of unfair pathways, ensuring a more precise handling of fairness issues and achieving a robust framework for mitigating unfairness.

## 5.3 Simplification of computing dividends

Unfortunately, as the size of the causal graph increases, removing the dividends of all unfair pathways and their unions using Lemma 5.1 can lead to an exponential increase in time complexity. However, the computational time can be significantly reduced. As shown in equation 7, it suffices to compute the sum of dividends over subsets $S \in \mathcal{F}_\pi$, corresponding to unfair pathways. We can rewrite it in a fairly simple form, as shown in the next theorem.

**Theorem 5.4.** *Let $P_{unfair}$ be a union of all unfair pathways, i.e., $P_{unfair} := \bigcup_{S \in \mathcal{F}_\pi} S$. Then it holds that*

$$f'(x) = f(x) + v(P_{unfair}) - v(\emptyset). \tag{9}$$

*Proof.* It is evident that $\sum_{S \in \mathcal{F}_\pi} \Delta_{v_{\mathcal{F}_\pi}}(S) = v_{\mathcal{F}_\pi}(P_{\text{unfair}})$. Then we have $\sum_{S \in \mathcal{F}_\pi \setminus \{\emptyset\}} \Delta_{v_{\mathcal{F}_\pi}}(S) = v_{\mathcal{F}_\pi}(P_{\text{unfair}}) - v_{\mathcal{F}_\pi}(\emptyset)$. Since $v_{\mathcal{F}_\pi}(S)$ returns $v(S)$ when $S \in \mathcal{F}_\pi$ and it is obvious that $P_{\text{unfair}} \in \mathcal{F}_\pi$, we have $f'(x) = f(x) + v(P_{\text{unfair}}) - v(\emptyset)$, and we complete the proof. $\square$

The key insight of Theorem 5.4 is that it is not necessary to treat the complex notions of feasible sets $\mathcal{F}$ and the dividends $\Delta_{v_{\mathcal{F}}}$ in the Möbius inversion formula. To proceed with our algorithm, it is sufficient to construct $P_{\text{unfair}}$ by the depth-first search and predict $v(\emptyset)$ and $v(P_{\text{unfair}})$.

**Example 3.** *Considering the example in Figure 1, we have $\mathcal{F}_\pi = \{\emptyset, \{A\}, \{A, D\}\}$, $P_{unfair} = \{A, D\}$. Consequently, $f'(x) = f(x) + v(\{A, D\}) - v(\emptyset)$.*

---

**Algorithm 1** Dividend Correction

---

1: **Input:** a causal graph $G$, a causal model, a predictive model $f$, a data point $x$, and unfair pathways $\pi$, a fairness parameter $\epsilon \in [0,1]$
2: **Output:** modified prediction $f'(x)$
3: Estimate the values of exogenous variables and train predictive models for each mediator.
4: Compute a reference value $r$ from the trained causal models.
5: Compute $P_{\text{unfair}}$, which is a union of all unfair pathways.
6: $f'(x) \leftarrow f(x) + \epsilon(v(P_{\text{unfair}}) - v(\emptyset))$.
7: **return** $f'(x)$.

---

### 5.4 Overall algorithm

The overall algorithm, called *dividend correction*, is given by Algorithm 1, which modifies the prediction such that the condition equation 7 is satisfied. A brief outline of the proposed algorithm is as follows: In line 3, it estimates the values of exogenous variables of each mediator and predicts causal models for them using the estimated values. In line 4, it creates a reference value $r$ on the basis of the causal model to build a set function $v$. The reference value of a sensitive attribute is considered 1, as the original value is 0. Features that are not affected by the sensitive attribute, e.g., $Q$, maintain their original values, whereas those affected by the sensitive attribute, e.g., $M$ and $D$, obtained their values from the trained causal models. In line 5, it computes a set $P_{\text{unfair}}$, used in Theorem 5.4, from unfair pathways. The set $P_{\text{unfair}}$ indicates the set of all nodes on unfair pathways and can be performed by proceeding a depth-first search in a causal graph $G$. In line 6, it eliminates the predictions of subsets computed in line 5 from $f(x)$ to produce a fair prediction $f'(x)$. A fairness parameter $\epsilon \in [0,1]$ is included to adjust the degree of fairness, where $\epsilon = 1$ indicates that it completely removes the effects of unfair pathways.

The computational complexity of our algorithm involves the time of the union procedure to compute $P_{\text{unfair}}$, the time to compute a reference value $r$, the time to estimate exogenous variables, and the time to learn causal models. The union procedure takes linear time based on the size of a DAG, as these are obtained by scanning a segment of the DAG consists of the unfair pathways. Let $d$ be the dimension of feature space and $g(d)$ be the time complexity of inference time by a causal model. The time complexity of computing the reference value takes $O(N \cdot g(d))$. Let $h(d)$ be the time complexity of computing a learning model. The time to compute causal models for mediators is $O(N \cdot h(d))$. Let $t$ be the iteration number of EM algorithm, and let $m$ be the number of data points. $e(m|N|t)$ denotes the time complexity of running EM algorithm.

**Theorem 5.5.** *The computational complexity of Algorithm 1 is $O(|E| + |N| \cdot (g(d) + h(d)) + e(m|N|t))$, where $N$ and $E$ are the set of nodes and edges in a DAG, respectively.*

## 6 Experiments

We demonstrate the effectiveness of the proposed algorithm by measuring the PSE, accuracy, and runtime using synthetic, Adult (Bache & Lichman, 2013), and German datasets. The experiments were performed using a Macbook Pro with Apple M1 Max and 64GB RAM.

### 6.1 Comparison with in-process approaches

We first evaluate the accuracy and PSE of our algorithm compared with some in-process approaches because there is no post-process approach for path-specific counterfactual fairness settings. Recall that instances for path-specific counterfactual fairness are parameterized by a causal graph $G$, a set of features $X$, and a model $f$ to predict $Y$. The detailed setups for each dataset are described in the appendix.

Our synthetic data setup was based on the configuration used by (Chikahara et al., 2021), which focused on a hiring decision scenario, as shown in Figure 1, indicating a causal graph $G$ with $X = \{A, D, Q, M\}$ and $A$ symbolizing sensitive attribute. We used unfair pathways $\pi = \{A \rightarrow Y, A \rightarrow D \rightarrow Y\}$ for the synthetic dataset.

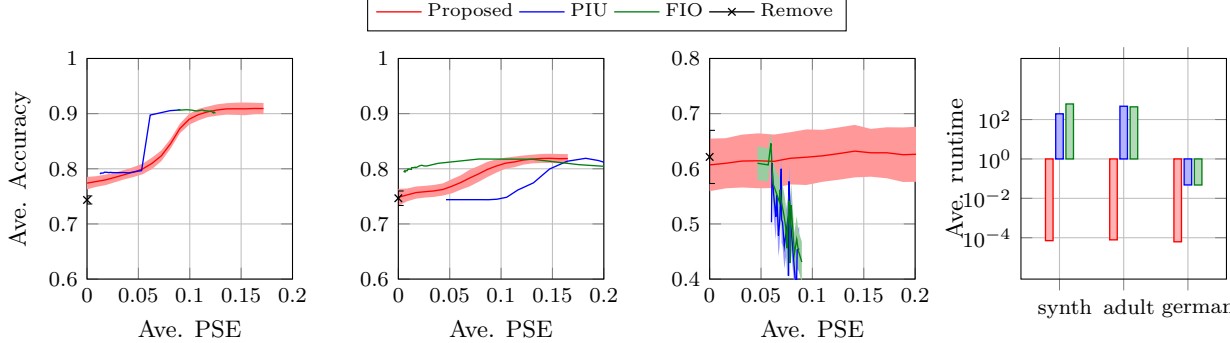

Figure 5: Test Accuracy vs. PSE for Synthetic (1st column), Adult (2nd column), and German datasets (3rd column), with runtime (in seconds) shown in the 4th column.

Our Adult setup was based on the configuration established by (Nabi & Shpitser, 2018; Chikahara et al., 2021), for which we used the source code provided on their Github repositories[4][5]. As shown in Figure 4, 5 attributes were examined: gender $A$, marital-status $M$, education $L$, occupation $R$, nationality $C$, from 11 attributes, because gender was identified as the sensitive attribute to be treated. A direct path from gender to income and indirect paths from gender to income via marital-status to unfair pathways were established, i.e., $\pi = \{A \to Y, A \to M \to Y, A \to M \to L \to Y, A \to M \to R \to Y, A \to M \to L \to R \to Y\}$.

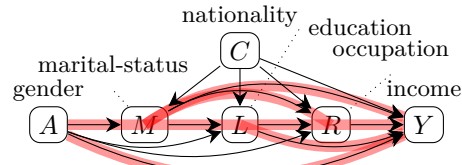

Figure 4: The causal graph of Adult dataset (Chikahara et al., 2021). The red pathways denote unfair pathways.

Our German [6] setting was based on (Chikahara et al., 2021). Nine attributes were examined: sex $A$, financial information $S$ (including saving accounts, checking account, and housing), information about debts $R$ (including credit amount and duration), and others $C$ (including age, job and purpose), as sex was defined as the sensitive attribute, as shown in Figure 8 of the appendix. We prepared three unfair paths from sex to each feature of financial information.

Given the absence of a post-processing approach specifically designed for path-specific counterfactual fairness, we carried out a comparative analysis of the proposed algorithms against three established in-process approaches: the probability of individual unfairness (PIU) (Chikahara et al., 2021; 2022), fair inference of outcome (FIO) (Nabi & Shpitser, 2018), and Remove algorithm. PIU aims to minimize the probability of individual unfairness, and FIO tries to reduce the mean unfair effect, defined in Appendix. To create the basic learning models for FIO and PIU, we employed logistic regression and neural network. In addition, we prepared Remove algorithm that learns a model without considering features that appear on unfair pathways. As learning models $f$, we implemented logistic regression (LR), random forest (RF), and neural network (NN) for our algorithm. Algorithms without fairness assessments can be directly implemented within our algorithm by setting $\epsilon = 0$. It is worth mentioning that thanks to its model-agnostic nature, our algorithm can adopt models where gradient-based learning does not work, such as RF.

Since the true causal models of mediators are unknown except for the synthetic dataset, we approximated them using linear regression. We estimated the values of exogenous variables that are not present in the original dataset. To this end, we utilized expectation-maximization (EM) algorithm (Dempster et al., 1977) to perform maximum likelihood estimation in latent variable models and to estimate the posterior distributions of the latent exogenous (noise) variables, as they are not directly observable. While the noise is assumed to

---

[4]https://github.com/raziehna/fair-inference-on-outcomes/

[5]https://github.com/ychika/IndividualLevel-PathSpecific-Counterfactual-Fairness/

[6]https://www.kaggle.com/datasets/uciml/german-credit

be additive, its specific values must still be estimated from the data [7]. With these estimated values, we then learned causal models for the mediators through linear regression. We applied these learning models as the true causal models when computing $Y_{A\Leftarrow 1\|\pi}$ in evaluating PSE. The effect of the model mismatch between the true causal model and learned causal model will be measured in Section 6.3.

Figure 5 shows the test accuracy and runtime of algorithms using neural networks as the base model, plotted against PSE for synthetic, Adult, and German datasets. The error regions represent one standard deviation. The results for other base models are given in Figure 11 in the Appendix. We varied the fairness-related parameters for each algorithm, except for the Remove algorithm. Each plot illustrates the average performance across 10 instances. The runtimes are averaged over the combinations of 10 instances and fairness parameters.

For Algorithm 1, we proceeded line 6, where $\epsilon$ alters from 0 to 1 in increments of 0.05. For both FIO and PIU, their penalty parameters were modified with respect to fairness from 0 to 2 in steps of 0.1. Figure 5 shows that superior results are characterized by higher accuracy and lower PSE. A notable result here is that our post-process approach is better than the existing in-process approaches when using logistic regression and remains competitive when using a neural network, despite in-process approaches having access to the causal graph during model training to achieve higher accuracy. Moreover, our algorithm achieves comparable accuracy while requiring significantly less runtime. Our algorithm could attain the better PSEs and runtime by utilizing the causal graph only in the post-process step but not during the model learning. The result with the highest PSE (the rightmost point) indicates the accuracy without any modification. By examining the gap between the accuracy at the highest PSE and the lowest PSE, one can assess the extent to which a learning model $f$ relies on unfair pathways. In the case of the Adult dataset, the random forest does not rely on unfair pathways to achieve higher accuracy compared to logistic regression and neural networks.

## 6.2 Comparison with post-process approaches

Since there are no existing post-processing approaches specifically targeting PSE, we compared the performance with other post-processing methods designed for CE. The performance was evaluated using the test accuracy and counterfactual effect (CE) on the synthetic dataset used in Section 6.2 and on the Adult dataset constructed in (Wu et al., 2019a), to compare the performance with other post-process approaches. The comparisons focus on post-process approaches, especially those for counterfactual fairness, which is a special case of path-specific counterfactual fairness. CE was computed using the PSE equation by setting all pathways from a sensitive attribute as unfair pathways. We leveraged the Adult dataset from their website[8], which consists of 7 features extracted from the original Adult dataset, shown in the appendix. We used unfair pathways $\pi = \{S \rightarrow I, S \rightarrow M \rightarrow I, S \rightarrow H \rightarrow I, S \rightarrow M \rightarrow W \rightarrow I, S \rightarrow M \rightarrow H \rightarrow I, S \rightarrow M \rightarrow W \rightarrow H \rightarrow I\}$, where $S, M, W, H, I$ denotes sex, marital-status, workclass, hours, and income, respectively. The causal graph is given in the Appendix.

We compared the proposed algorithm with algorithms suggested by (Wu et al., 2019a) denoted by CF to assess its performance and used LR and SVM as baseline models for post-process approaches. Similar to experiments on in-process approaches, linear regression was used as a causal model of mediators.

Figure 6 presents the CE and accuracy, which includes error regions averaged across 20 instances for the synthetic and Adult datasets. The runtime can be found in Figure 12 of the Appendix. The same fairness parameter settings were used for the proposed algorithms as in the experiments in Section 6.1. The fairness parameter of the CF was varied from 0 to 0.2 in steps of 0.01. Our algorithm with SVM outperforms other existing algorithms on the synthetic dataset, and it also achieves superior performance with both LR and SVM outperform on Adult dataset. Although our algorithm with LR on the synthetic dataset is an exception, it always achieves a better CE, demonstrating its effectiveness in improving fairness.

---

[7]Other algorithms for estimating posterior distributions, such as Gaussian approximation, can also be applied to our settings, as demonstrated in (Chiappa, 2019)

[8]https://www.yongkaiwu.com/publication/

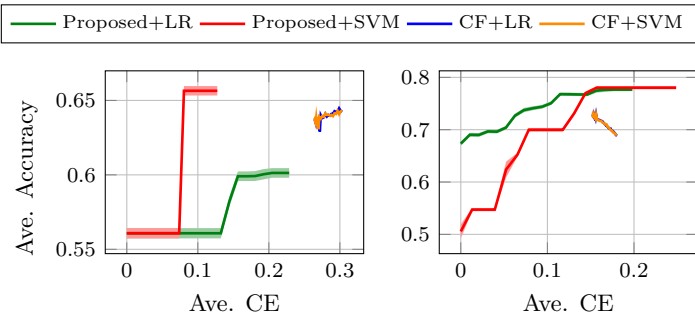
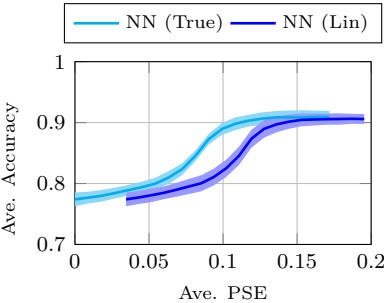

Figure 6: Averaged test accuracy v.s. averaged CE on synthetic(left) and the Adult datasets(right).

Figure 7: Accuracy v.s. PSE on the misspecification setting.

## 6.3 Experiment on model mismatch

In Section 6.1, we assume trained predictive models for mediators to be true. In this section, we evaluate the performance differences between scenarios where the true causal models of mediators are known and where they are trained models. To this end, we adopted true causal models as mediator models when modifying $Y_{A \Leftarrow 0}$, meaning that our algorithm employs the trained model instead of true model when calculating a reference value $r$. We then verified $Y_{A \Leftarrow 1 \| \pi}$ using the true causal model. Figure 7 shows the average accuracy against the PSE across 10 instances. We evaluated the proposed algorithm using neural network as a base learning model. The results for the other base models can be found in Figure 13. The labels (True) and (Lin) signify that the algorithm uses the true causal model and linear regression, respectively, in the computation of a reference value $r$. These results indicate that the model mismatch had no significant impact on accuracy and only a slight effect on PSE, up to 0.05. Furthermore, it is evident that using more precise mediator models would lead to a smaller disparity.

## 7 Conclusion

We propose a new post-processing method to achieve path-specific counterfactual fairness. Our method is based on a set function representation of a machine learning model. By applying the Möbius inversion formula, we prove that every set function can be represented as the sum of causal effects discussed previously. To elucidate the relationship between the causal effects and the path-specific counterfactual fairness, we propose a causally aware set function by incorporating feasible sets and interior operator. Despite introducing the aforementioned complex concepts, the resulting algorithm is surprisingly simple and empirically works efficiently. Furthermore, our algorithm is the first post-processing method that efficiently achieves path-specific counterfactual fairness and it is a model-agnostic method that can handle any model architecture and feature representation.

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

# Appendix

## A    Broader Impact Statement

Using incorrect or inaccurate causal graphs or selecting wrong unfair pathways can result in discriminatory effects, which in turn may have negative societal impacts, as highlighted by our research. A limitation of the proposed method is that it ensures fairness within the context of the provided causal graph, although the validity of the causal graph itself remains a separate concern, beyond the scope of our method. Therefore, it is crucial to continually validate and update the causal graph's validity through expert assessment. A notable advantage of the post-processing approach is that it does not require model re-training when the causal graph is updated.

## B    Examples

**Example 4.** *Having Example 1, we shall compute $\Delta_{v_{\mathcal{F}}}(\{A, Q, D\})$. Suppose we have a set function $v$ with $v(\emptyset) = 0$, $v(\{A\}) = 10$, $v(\{Q\}) = 20$, $v(\{D\}) = 30$, $v(\{A, Q\}) = 40$, $v(\{A, D\}) = 30$, $v(\{Q, D\}) = 50$, and $v(\{A, Q, D\}) = 100$. The dividend of $\{A, Q, D\}$ on $v_{\mathcal{F}}$ is given by $\Delta_{v_{\mathcal{F}}}(\{A, Q, D\}) = v(\{A, Q, D\}) - v(\{A, Q\}) = 60$ because $en(\{A, Q, D\}) = \{D\}$.*

Lemma 5.3 can be easily understood by using linear causal models.

**Example 5.** *In a simple case where models $f$, $f_D$, and $f_M$ are linear, the PSE of Figure 1, when $A$ flips from $a$ to $a'$, can be computed as follows. We can represent the models $f$, $f_D$, and $f_M$ as $f_D(A, Q) = c^d + c_a^d A + c_q^d Q + \epsilon_d$, $f_M(A, Q) = c^m + c_a^m A + c_q^m Q + \epsilon_m$, and $f(A, Q, D, M) = c^y + c_a^y A + c_q^y Q + c_d^y D + c_m^y M + \epsilon_y$, where $A \in \{a, a'\}$, $Q = \epsilon_q$, and $c^d, c_a^d, c_q^d, c^m, c_a^m, c_q^m, c^y, c_a^y, c_q^y, c_d^y, c_m^y, \epsilon_d, \epsilon_m, \epsilon_y \in \mathbb{R}$. When we set $\pi = \{A \to Y, A \to D \to Y\}$ as unfair pathways, its PSE is given by $c_a^y(a' - a) + c_d^y c_a^d(a' - a)$. On the other hand, dividends $\Delta_{v_{\mathcal{F}}}(\{A\})$ and $\Delta_{v_{\mathcal{F}}}(\{A, D\})$ are given by $\Delta_{v_{\mathcal{F}}}(\{A\}) = v_{\mathcal{F}}(\{A\}) - v_{\mathcal{F}}(\emptyset) = f(a, q', d', m') - f(a', q', d', m') = c_a^y(a - a')$ and $\Delta_{v_{\mathcal{F}}}(\{A, D\}) = v_{\mathcal{F}}(\{A, D\}) - v_{\mathcal{F}}(\{A\}) - v_{\mathcal{F}}(\{D\}) + v_{\mathcal{F}}(\emptyset) = f(a, q', d, m') - f(a, q', d', m') = c_d^y c_a^d(a - a')$, where $v_{\mathcal{F}}(\{D\}) = v_{\mathcal{F}}(\emptyset)$. Hence, the PSE can be rephrased as $-\Delta_{v_{\mathcal{F}}}(\{A\}) - \Delta_{v_{\mathcal{F}}}(\{A, D\})$.*

**Example 6.** *Feasible sets on unfair pathways $\mathcal{F}_\pi$ of Example 1 is given by $\mathcal{F}_\pi = \{\emptyset, \{A\}, \{A, D\}\}$.*

## C    Detailed experimental settings

### C.1    Details of baseline algorithms

PIU (Chikahara et al., 2022) aims to minimize the *probability of individual unfairness*, which is defined by

$$P(Y_{A \Leftarrow 1 \| \pi} \neq Y_{A \Leftarrow 0}).$$

FIO (Nabi & Shpitser, 2018) seeks to minimize the *mean unfair effect*, defined by

$$\mathbb{E}_{Y_{A \Leftarrow 1 \| \pi}, Y_{A \Leftarrow 0}}[Y_{A \Leftarrow 1 \| \pi} - Y_{A \Leftarrow 0}] = P(Y_{A \Leftarrow 1 \| \pi} = 1) - P(Y_{A \Leftarrow 0} = 1).$$

Remove algorithm learns a predictive model using base models (logistic regression and neural network) while excluding features associated with unfair pathways.

### C.2    Details of datasets on comparison with in-process approaches

**synthetic**    Our synthetic data setup was based on the configuration used by (Chikahara et al., 2021), which focused on a hiring decision scenario, as shown in Figure 1, indicating a causal graph $G$ with $X = \{A, D, Q, M\}$ and $A$ symbolizing sensitive attribute. The values of $A, D, Q,$ and $M$ were sampled from the

following causal model:

$$A = U_A, \quad U_A \sim \text{Bernoulli}(0.6),$$
$$Q = \lfloor U_Q \rfloor, \quad U_Q \sim \mathcal{U}(0.1, 1),$$
$$D = A + \lfloor 0.5QU_D \rfloor, \quad U_D \sim \text{Tr}\mathcal{N}(2, 1^2, 0.1, 3.0),$$
$$M = 3A + 0.4QU_M, \quad U_M \sim \text{Tr}\mathcal{N}(3, 2^2, 0.1, 3.0),$$
$$Y = \text{Bernoulli}(\varsigma(-10 + 5A + Q + D + M)),$$

where Bernoulli, $\mathcal{U}$, and Tr$\mathcal{N}$ denotes the Bernoulli, uniform, and truncated Gaussian distributions, respectively, and $\varsigma(x) := 1/(1 + \exp(-x))$ is a standard sigmoid function. We generated 10 instances, each of which is sampled 6000 data points and separated them into 5000 as training data and 1000 as test data. We used unfair pathways $\pi = \{A \to Y, A \to D \to Y\}$ for the synthetic dataset.

**Adult** Our experimental setup was based on the configuration established by (Nabi & Shpitser, 2018; Chikahara et al., 2021), for which we used the source code provided on their Github repositories[9][10]. The adult dataset is commonly used for predicting an individual's income level by determining if their salary is above or below \$50,000. Five attributes were examined: gender $A$, marital-status $M$, education $L$, occupation $R$, nationality $C$, from 11 attributes as shown in Figure 4, as gender was identified as the sensitive attribute to be treated. To construct the datasets, 5000 records were selected for training and 1000 records for testing from a total of 65,123 records. A direct path from gender to income and indirect paths from gender to income via marital-status to unfair pathways were established, i.e., $\pi = \{A \to Y, A \to M \to Y, A \to M \to L \to Y, A \to M \to R \to Y, A \to M \to L \to R \to Y\}$. Given unfair pathways, $P_{\text{unfair}}$ is given by $\{A, M, L, R\}$.

**German** Our german data setting was based on (Chikahara et al., 2021). Nine attributes were examined: sex $A$, financial information $S$ (including saving accounts, checking account, and housing), information about debts $R$ (including credit amount and duration), and others $C$(including age, job and purpose), as sex was defined as the sensitive attribute, as shown in Figure 8. We prepared three unfair paths from sex to each feature of financial information. To construct the dataset, we selected 900 records for training data and 100 records for testing from a total of 1000 records. Unfair pathways are set to $\pi = \{A \to Y, A \to S \to Y\}$. Given unfair pathways, we have $P_{\text{unfair}} = \{\text{sex}, \text{including saving accounts}, \text{checking account}, \text{housing}\}$.

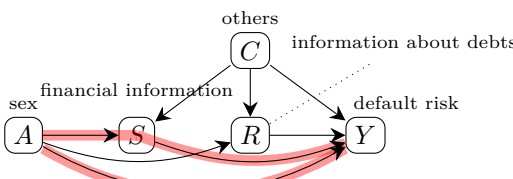

Figure 8: The causal graph of German setting (Chikahara et al., 2022) for comparison with in-process approach. The red paths from the sensitive attribute $S$ to $Y$ denote unfair paths.

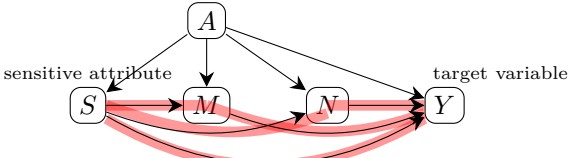

Figure 9: The causal graph of the synthetic dataset (Wu et al., 2019a) for comparison with post-process approach. The red paths from the sensitive attribute $S$ to $Y$ denote unfair paths.

---

[9]https://github.com/raziehna/fair-inference-on-outcomes/
[10]https://github.com/ychika/IndividualLevel-PathSpecific-Counterfactual-Fairness/

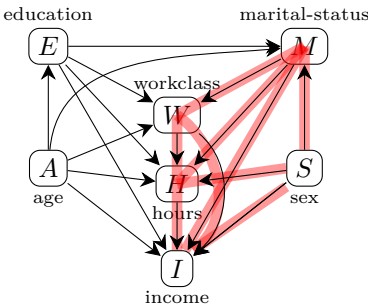

Figure 10: The causal graph of Adult dataset (Wu et al., 2019a) for comparison with post process approaches. The red paths from the sensitive attribute $S$ to $I$ denote unfair paths.

### C.3  Estimation of exogenous variables

We estimated the values of exogenous variables through the EM algorithm. We setup the expectation and maximization steps as the following: (i) In the expectation step, at each iteration, we use the current parameter estimates to compute the expected value of the exogenous variable given the observed data $(X, y)$, expressed as:

$$Z = \frac{y - X\beta}{\gamma},$$

where $Z$ represents the estimated exogenous variable, $\beta$ is the coefficient parameter for a linear model, and $\gamma$ is a noise parameter. (ii) In the maximization step, we update the parameters $\beta$ and $\gamma$ as follows. We find the best fit for $\beta$ using the least square method. As for updating $\gamma$, we compute it with the following formula:

$$\gamma = \frac{\sum (y - X\beta)Z}{\|Z\|_2^2}.$$

The expectation and maximization steps are iterated $T$ times. In our experiments, for each mediator, we utilized all its observable parent attributes for $X$ and set the value of the mediator as $y$, and we set $T = 100$ for the number of iterations of the EM algorithm.

### C.4  Additional comparisons with in-process approaches

Figure 11 shows the test accuracy and runtime of algorithms using logistic regression, random forest, and neural networks as the base model, plotted against PSE for synthetic, Adult, and German datasets. The error bars representing the standard deviation. We varied the fairness-related parameters for each algorithm, except for the Remove algorithm. Each plot illustrates the average performance across 10 instances. The runtimes are averaged over the combinations of 10 instances and fairness parameters.

### C.5  Comparisons with post-process approach

we measured accuracy and counterfactual effect (CE) on the synthetic dataset used in Section 6.2 and on the adult dataset used in (Wu et al., 2019a). For both the synthetic and Adult datasets, we utilized dataset that was made available on their website[11]. The causal graph of the synthetic data is shown in Figure 9. Each entry of the Adult dataset includes 7 features extracted from the original adult dataset as shown in Figure 10. We set direct path from sex to income and indirected paths from sex to income via marital-status are set to unfair pathways, that is, $\pi = \{S \to I, S \to M \to I, S \to H \to I, S \to M \to W \to I, S \to M \to H \to I, S \to M \to W \to H \to I\}$, where $S, M, W, H, I$ denotes sex, marital-status, workclass, hours, and income, respectively. With the unfair pathway $\pi$, we have $P_{\text{unfair}} = \{S, M, H, W\}$. We have 65,123 data points and divided them into an 80/20 training and test dataset split.

---

[11] https://www.yongkaiwu.com/publication/

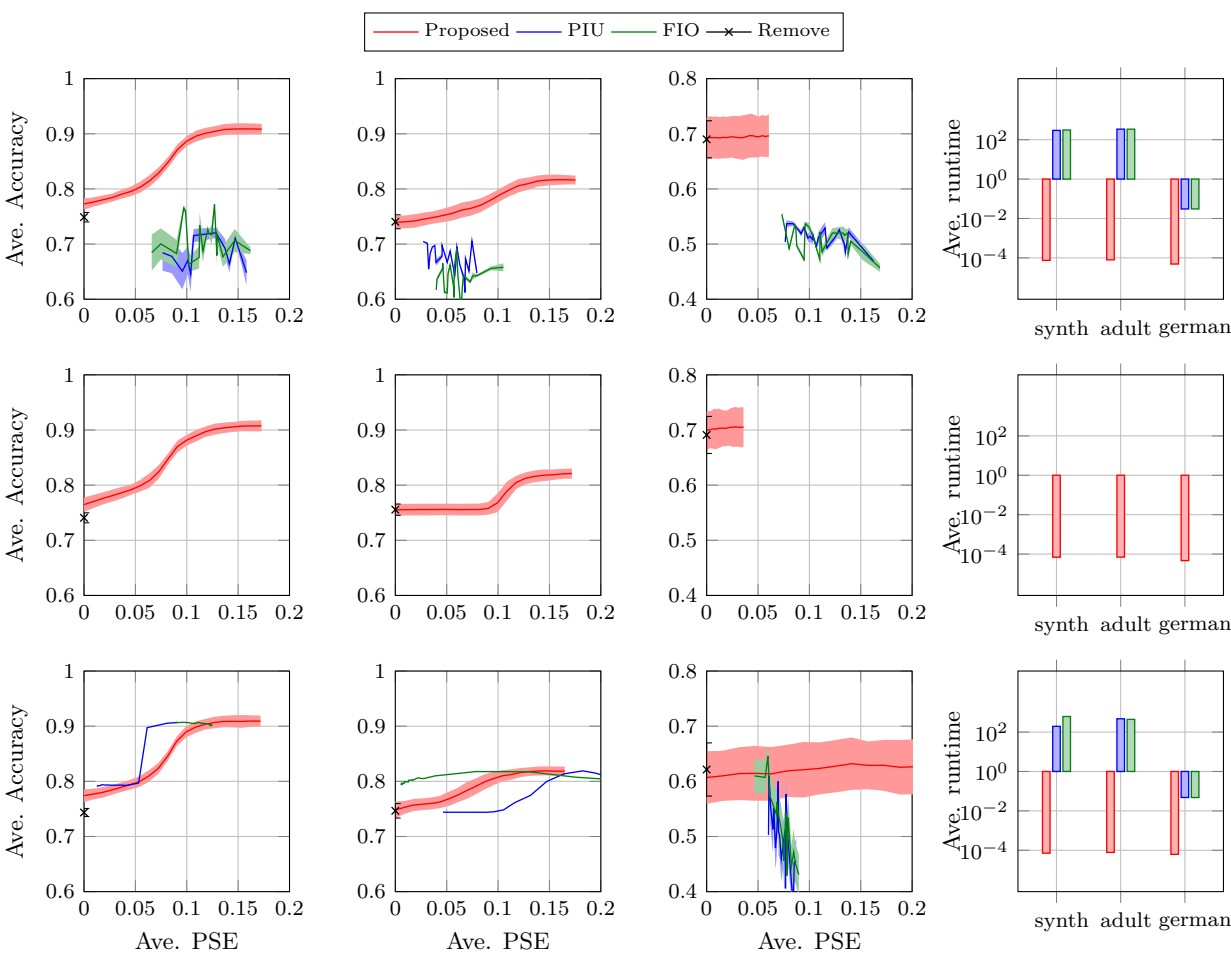

Figure 11: Test Accuracy vs. PSE for Synthetic (1st column), Adult (2nd column), and German datasets (3rd column), with runtime (in seconds) shown in the 4th column. Rows represent base model: Logistic Regression (1st row), Random Forest (2nd row), and Neural Network (3rd row). Error bars represent standard deviation. In the middle row, the results of PIU and FIO do not appear because they cannot use random forest as a base model.

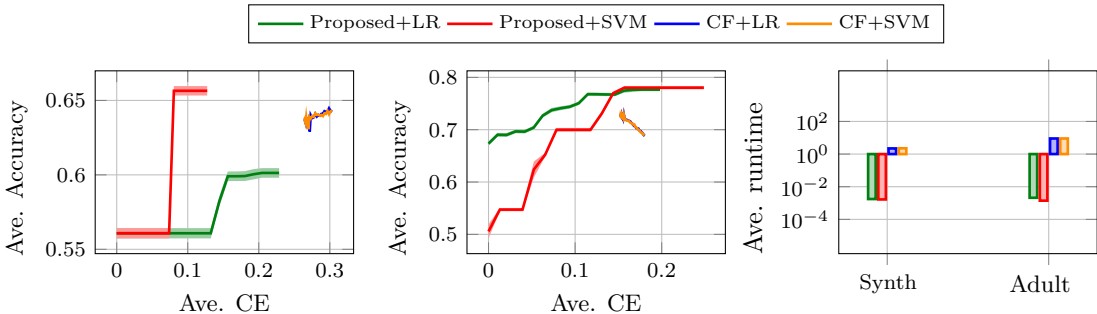

Figure 12: Averaged test accuracy v.s. averaged CE on the synthetic(left) and Adult datasets(center) and runtime(right).

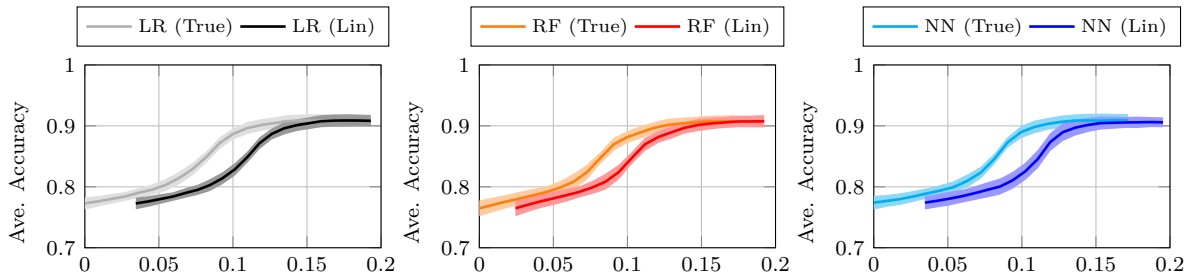

Figure 13: Accuracy v.s. PSE on the misspecification setting.

**Baselines & Proposed algorithm** We compared proposed algorithm with algorithms proposed by (Wu et al., 2019a) denoted by CF. We utilized logistic regression (LR) and support vector machine (SVM) as baseline models of post-process approaches.

**Result** Figure 6 plots the test accuracy against CE and averaged runtime when changing the parameters regarding fairness on the synthetic and the adult datasets. We used the same settings as those in the comparisons for in-process approaches in Section 6.1, as per the proposed algorithm. As for CF, we altered a fairness parameter $\tau$ from 0.01 to 0.3 in steps of 0.01. When we compute CE, i.e., $P(Y_{S\Leftarrow 1} \mid z) - P(Y_{S\Leftarrow 0} \mid z)$, for both synthetic and adult dataset and for both proposed algorithms and CF, we utilized a linear regression models learned from data to calculate the value of each mediator in changing the sensitive attribute value to compute the first term. For each parameter setting, we averaged CE over all data points and plots them in Figure 6, and averaged runtime over all combinations of instances and fairness parameters. We can observe that in the adult dataset, our algorithms clearly outperform existing algorithms. Conversely, our algorithm with support vector machine outperforms existing algorithms, while the algorithm with logistic regression achieves lower CE but also lower accuracy than existing one. Furthermore, our algorithm significantly improve the runtime compared with the existing algorithm.

### C.6 Additional experiments on model mismatch

Figure 13 shows the average accuracy against the PSE across 10 instances. We evaluated the proposed algorithm using logistic regression(LR), random forest(RF), and neural network(NN) as a base learning model. The labels (True) and (Lin) signify that the algorithm uses the true causal model and linear regression, respectively, in the computation of reference value $r$. These results indicate that the model mismatch had no significant impact on accuracy and only a slight effect on PSE, up to 0.05. Furthermore, it is evident that using more precise mediator models would lead to a smaller disparity.

## D   Proof of Lemma 5.1

**Lemma 5.1.** *Let $\mathcal{F}_\pi$ be a feasible set of a DAG induced by unfair pathways $\pi$. Then the prediction $f'(x)$ is path-specific counterfactual fair if and only if it holds that*

$$\sum_{S \in \mathcal{F}_\pi \setminus \{\emptyset\}} \Delta_{v_{\mathcal{F}_\pi}}(S) = 0. \tag{10}$$

*Proof.* In this proof, we will write a set function $v$ given a data point $x$ and a reference value $r$ on feasible set $\mathcal{F}$ as $v_{x,r,\mathcal{F}}$. Let $N_{\pi^c}$ be a set of nodes excluded in unfair pathways $\pi$, and let $v_{x,r'}$ be a causally-aware set function that considers only the effects of unfair pathways $\pi$ for any $v$. The $i$-th element of the reference value $r'$ holds $r'_i = r_i$ if $i$ is on unfair pathways and $r'_i = x_i$ otherwise, where $r$ is a reference value constructed in Section 5.1. The resulting set function $v_{x,r'}$ holds $v_{x,r',\mathcal{F}}(S \setminus T) = v_{x,r,\mathcal{F}}(S)$ for all $S \subseteq N$ and $T \subseteq S \cap N_{\pi^c}$. $Y_{A \Leftarrow 1 \| \pi}$ and $Y_{A \Leftarrow 0}$ can be written by using $v_{x,r'}$. For every $S \in \mathcal{F}$, the dividend of $v_{x,r'}$ can be represented by that of $v_{x,r}$, given by

$$\begin{aligned}
\Delta_{v_{x,r',\mathcal{F}}}(S) &= \sum_{T \subseteq en(S)} (-1)^{s-t} v_{x,r',\mathcal{F}}(S \setminus T) \\
&= \sum_{T \subseteq en(S) \setminus N_{\pi^c}} \sum_{W \subseteq en(S) \cap N_{\pi^c}} (-1)^{s-(t'+w)} v_{x,r',\mathcal{F}}(S \setminus (T \cup W)) \\
&= \sum_{T \subseteq en(S) \setminus N_{\pi^c}} \sum_{W \subseteq en(S) \cap N_{\pi^c}} (-1)^{s-(t+w)} v_{x,r,\mathcal{F}}(S \setminus T) \\
&= \sum_{W \subseteq en(S) \cap N_{\pi^c}} (-1)^{w} \sum_{T \subseteq en(S) \setminus N_{\pi^c}} (-1)^{s-t} v_{x,r,\mathcal{F}}(S \setminus T) \\
&= (1-1)^{|en(S) \cap N_{\pi^c}|} \sum_{T \subseteq en(S) \setminus N_{\pi^c}} (-1)^{s-t} v_{x,r,\mathcal{F}}(S \setminus T) \\
&= \begin{cases} \sum_{T \subseteq en(S) \setminus N_{\pi^c}} (-1)^{s-t} v_{x,r,\mathcal{F}}(S \setminus T) & \text{if } |en(S) \cap N_{\pi^c}| = 0, \\ 0 & \text{otherwise.} \end{cases} \\
&= \begin{cases} v_{x,r,\mathcal{F}}(S \setminus N_{\pi^c}) & \text{if } |en(S) \cap N_{\pi^c}| = 0, \\ 0 & \text{otherwise,} \end{cases}
\end{aligned}$$

where $s, t, w$ be the size of $S, T, W$, respectively, the third equality follows from the property of $v_{x,r',\mathcal{F}}$, and the fifth equality follows from the binomial theorem. The equation $|en(S) \cap N_{\pi^c}| = 0$ indicates that no feature in $N_{\pi^c}$ is not the bottom of DAG induced by $S$, and the above equation shows that $S \setminus N_{\pi^c}$ is a feasible set on $v_{x,r',\mathcal{F}}$ instead of $S$. This implies that feasible sets of $\hat{v}$ are equivalent to that of a DAG induced by unfair pathways $\pi$, i.e., $\mathcal{F}_\pi$. Then we have

$$\sum_{S \in \mathcal{F}} \Delta_{v_{x,r',\mathcal{F}}}(S) = \sum_{S \in \mathcal{F}_\pi} \Delta_{v_{x,r,\mathcal{F}_\pi}}(S).$$

By regarding a given data point $x$ as a reference value and the data whose sensitive attribute's value changed to 1 as a baseline value, we have

$$\begin{aligned}
\mathbb{E}_{Y_{A \Leftarrow 1 \| \pi}, Y_{A \Leftarrow 0}}[Y_{A \Leftarrow 1 \| \pi} - Y_{A \Leftarrow 0} \mid X = x] &= v_{x,r',\mathcal{F}}(N) - v_{x,r,\mathcal{F}}(\emptyset) \\
&= \sum_{S \in \mathcal{F}} \Delta_{v_{x,r',\mathcal{F}}}(S) - \Delta_{v_{x,r,\mathcal{F}}}(\emptyset) \\
&= \sum_{S \in \mathcal{F}_\pi} \Delta_{v_{x,r,\mathcal{F}_\pi}}(S) - \Delta_{v_{x,r,\mathcal{F}}}(\emptyset) \\
&= \sum_{S \in \mathcal{F}_\pi \setminus \{\emptyset\}} \Delta_{v_{x,r,\mathcal{F}_\pi}}(S),
\end{aligned}$$

where the last equality follows from that it always holds $\emptyset \in \mathcal{F}, \mathcal{F}_\pi$. Hence, the proof is complete. $\qquad\square$

# E   Proof of Proposition 5.3

**Proposition 5.3.** *A dividend $-\Delta_{v_{\mathcal{F}}}(S)$ shows the direct causal effect if $S$ is a root node of $G$, the indirect causal effect if $S$ is a pathway, and the effect of interaction if $S$ is combined with multiple pathways.*

*Proof.* Without loss of generality, let us consider a set of nodes denoted as $N$, with the top-most node being $A$ and the other nodes can be reached from $A$[12]. This implies that the effect of $A$ influences all other nodes within $N \setminus \{A\}$.

Every feasible set conforms to one of the following structural types: a singleton, a pathway and the combination of pathways. We prove that $\Delta_{v_{\mathcal{F}}}$ for a singleton, a pathway, and the combination of pathways align with total direct effect, path-specific effect, and mediated interaction, respectively. Recall that the total effect by altering the sensitive attribute can be decomposed into the total direct effect, path-specific effects, and mediated interactions, as defined in Section 3.2.

We begin by computing the direct causal effect $A \to Y$, i.e., $S = \{A\}$. In path-specific counterfactual fairness settings, as only a sensitive attribute $A$ is intervened, we treat a direct causal effect $A \to Y$. It is evident that $-\Delta_{v_{\mathcal{F}}}(\{A\})$ represents the total direct effect, as shown by

$$-\Delta_{v_{\mathcal{F}}}(\{A\}) = v_{\mathcal{F}}(\emptyset) - v_{\mathcal{F}}(\{A\}).$$

This equation implies that $-\Delta_{v_{\mathcal{F}}}(\{A\})$ is the difference between the predicted value when all features changes due to the change in $A$ and the predicted value when all features except for $A$ change. This computation is equivalent to the total direct effect. For the other features $i \in N \setminus A$, we have

$$-\Delta_{v_{\mathcal{F}}}(\{i\}) = v_{\mathcal{F}}(\emptyset) - v_{\mathcal{F}}(\{i\}) = 0,$$

where $v_{\mathcal{F}}(\{i\}) = v_{\mathcal{F}}(\emptyset)$ since $i$ is not the top-most node. In other words, $i$ does not exist in a feasible set $\mathcal{F}$ as a singleton, that is, $\{i\} \notin \mathcal{F}$.

Next, we show the computation about path-specific effects. Recall that the path-specific effect can be computed as the difference between the predicted value when the mediator takes on a specific value influenced by the treatment that the sensitive attribute changes from 0 to 1 and the predicted value when the value of mediator remain unchanged. In our set function representation, every pathway $S$ with $|S| \geq 2$ has a characteristic that $S$ has a unique single node $i$ that do not have any child, that is $en(S) = \{i\}$. This indicates that a node $i$ is the last mediator on the pathway $S$. In this case, the dividend can be computed as follows.

$$-\Delta_{v_{\mathcal{F}}}(S) = v_{\mathcal{F}}(S \setminus i) - v_{\mathcal{F}}(S).$$

This equation implies that $-\Delta_{v_{\mathcal{F}}}(S)$ is the difference between the predicted value when all features except for $i$ remain unchanged but the feature $i$ takes the value influenced by the change of the sensitive attribute. As a node $i$ works as a mediator, this computation corresponds to the path-specific effect.

Finally, we provide the computation about the mediated interaction. Recall that the mediated interaction among mediators can be computed by taking the sum of the predicted value when changing the value of mediators but their signs alternate based on the number of changed mediators. In our set function representation, the combination of pathways, e.g., $S$, has a characteristic that $|en(S)| \geq 2$, which can be seen as the number of mediators that do not have any child. Thus, we have

$$-\Delta_{v_{\mathcal{F}}}(S) = - \sum_{T \subseteq en(S)} (-1)^{|T|} v_{\mathcal{F}}(S \setminus T).$$

This implies that the dividend can be computed as the sum of predicted value $v_{\mathcal{F}}(S \setminus T)$, where $T$ is the set of mediators that changes its value influenced by the alternation of the sensitive attribute and its sign differs based on the number of changed mediators $T$. This computation aligns with the mediated interaction.

$\square$

---

[12]In cases where the causal graph does not have single top-most node, the same problem setting can be achieved by assigning the same values to the nodes $T$ that are not influenced by $A$, i.e., $x_T = r_T$.

