# OpenReview forum: "Path-Specific Counterfactual Fairness via Dividend Correction"
_TMLR — Accepted by TMLR_

### Review · Reviewer_Ntd9 · 2024-12-06

**Summary Of Contributions:**

The authors present a novel post processing method for achieving path-specific counterfactual fairness for general machine learning models. To this end, the authors leverage the Möbius inversion formula and a novel notion called *causally aware set functions* to derive a characterization for path-specific counterfactual fairness. From this characterization, the authors derive a five-step algorithm to *''correct''* any machine learning model, for which the underlying causal graph is provided. Experimental evaluations on synthetic and real world data demonstrate the merits of the proposed method in comparison to prior work.

**Audience:**

Yes

**Broader Impact Concerns:**

No broader impact concerns.

**Claims And Evidence:**

Yes

**Requested Changes:**

Crucial requested changes are highlighted by the term *(crucial)*.

**Abstract**

- *By means of the mathematical tools in cooperative game* This seems to be a typo. I believe the authors mean something like *By means of the mathematical tools from cooperative game theory*.

**Introduction**

- *Their framework assumes that any causal effect of the sensitive attribute on the
outcome as inherently problematic and aims to eliminate any influence that the sensitive attribute might
have on the outcome.* I do not understand this sentence. What does *problematic* mean? If we assume that no attributes should be causal for the prediction (we eliminate the influences), we end up without any attributes at all. Disclaimer: I am not very familiar with the framework by [1].

- *we introduce the Möbius inversion formula* It does not seem like the authors introduce the formula. It seems instead that they introduce this formula to the field of counterfactual fairness. Please clarify.

- *We newly introduce a novel causally aware set function* Typo. Please remove *newly*.

**Related Work**

- *n-processing approach adapts the learning algorithm...* Typo. Please write *The n-processing approach adapts the learning algorithm...*

- *n-process approaches are tend to be more complex* Typo. Please remove *are*.

- *Our finding will provide a new possible connection between machine learning and cooperative game* This seems to be the same typo as in the abstract. Please fix or clarify.

- *in the analyze of causality.* Typo. Please write *in the analysis of causality*.

**3.1 Path-specific counterfactual fairness**

- *Since we focus on the impact of the sensitive attribute, we only
consider exogenous variables that are not influenced by the sensitive attribute.* I am a bit confused by this sentence. From my understanding, exogenous variables are never influenced by endogenous variables. Please clarify.

- I do not understand why $Y_{A \Leftarrow 0}$ does not use unfair pathways. Why is it not $Y_{A \Leftarrow 0 || \pi}$? The unfair mechanisms are also used here.

**3.2 A decomposition of the total effect into path-specific effects**

- In the definition of total effect of alternation (TE), I found it hard to see what the difference to the compared variables in the definition of path-specific counterfactual fairness (Definition 3.1) is. I believe $Y_{A \Leftarrow 1 || \pi}$ means that you set $A = 1$ *only* in the unfair pathways and here, you set $A = 1$ for all of pathways. It would be nice to elevate this a bit more.

- *... showed that the sum of path-specific effects does not equivalent to the total effect*. Typo. Please write *... showed that the sum of path-specific effects is not equivalent to the total effect*.

**4 Problem setting and motivating scenario**

- *the publisher must publicly disclose the causal graph $G$ that underpins the predictive model $f$*. I am not very familiar with counterfactual fairness. However, it seems to me that most publishers simply fit their model to data without imposing a causal structure. That means they will not be able to provide such a causal graph. In this case, it seems methods such as the proposed one cannot be applied (?).

- (crucial) *This is important because, when the publisher can define their causal models, the values of mediators can be adjusted as desired when altering the sensitive attribute* I do not understand this sentence. Unfortunately, it seems relevant, because it seems to be to motivation for the proposed method. Please add more explanation here, maybe using the concrete example from section 3.1.

**5.1 Construction of causally aware set function**

- The section title has a typo. Please write instead *5.1 Construction of causally aware set functions*.

- *These conditions are necessarily...* Typo. Please replace by *These conditions are necessary*.

- *must either be measured and accounted for* Typo. Please replace by *must either be measured or accounted for*.

- Not a requested change, but I have a feeling that this interior operator could be related to backtracking counterfactuals ([2, 3]), though I may be wrong. The similarity seems that analagously to this operator, backtracking counterfactuals require that causal parent variables of antecedent variables are taken into account.

**5.2 Decomposition into the sum of causal effects**

- (crucial) Equation 6: *Intuitively, $\Delta_{v_\mathcal{F}}(T)$ indicates that the synergy effect of combining with T features* This sentence makes no sense (wrong grammar and no content) and I believe it would make sense to spend much more text on providing the reader with an intuition for this variable. I find this variable very hard to grasp just like that.

**6.Experiments**

- *and German datasets* The reference for the data set is missing.

- *To this end, we utilized expectation-maximization (EM) algorithm* I do not understand why the EM algorithm is required to infer the noise variables if we assume additive noise. Specifically, $e_i = x_i - f_i(PA(x_i))$. Or are the latent variables for the EM algorithm not the noise variables, but some model parameters? If yes, it would be interesting to know what latent variable model was fit here.

- *which includes error bars averaged across 20 instances for the synthetic and Adult datasets* Do these error bars correspond to 1 std.? If yes, please add that for clarity. Otherwise, please write what exactly the bars correspond to.


[1] Kusner, Matt J., et al. "Counterfactual fairness." Advances in neural information processing systems 30 (2017).

[2] Von Kügelgen, Julius, Abdirisak Mohamed, and Sander Beckers. "Backtracking counterfactuals." Conference on Causal Learning and Reasoning. PMLR, 2023.

[3] Kladny, Klaus-Rudolf, et al. "Deep backtracking counterfactuals for causally compliant explanations." Transactions on Machine Learning Research (2024).

**Strengths And Weaknesses:**

**Strengths**

1. The findings seem to present a significant contribution to the field. The proposed method seems quite non-trivial, yet the resulting algorithm is incredibly simple.

2. The authors did an excellent job in terms in regards to presentation (apart from grammar mistakes, see weakness 2). Generally, the manuscript provides intuitive explanations and illustrative examples. These render the manuscript accessible to a broad audience.

3. The method is agnostic of the used machine learning model.

4. Many experiments on synthetic and real world data demonstrate the merits of the proposed method empirically.

**Weaknesses**

1. The proposed method only works for binary sensitive attributes $A$. It is easy to see that this assumption is restrictive, because many variables that lead to discrimination are not binary (e.g., ethnicity or age).

2. There are quite a lot of grammar mistakes in the manuscript that are irritating. It should take little time to fix them and it will greatly enhance the reading experience.

3. The theoretical preliminaries for understanding this method are quite complicated and I believe that much more effort should be spent on explaining some parts. Especially, I have a hard time understanding what this *Möbius transformer* really is and it would be great if more words could be spent on this.

4. Maybe I missed it, but I could not find source code for reproducibility of the experiments.

---

> ### Author Response · Authors · 2025-01-15
> **Response to Reviewer Ntd9**
>
> Thank you for giving your insightfull comments and for pointing out the typos. We really appreciate your attention to detail and your help in improving our manuscript.
>
> > Their framework assumes that any causal effect of the sensitive attribute on the outcome as inherently problematic and aims to eliminate any influence that the sensitive attribute might have on the outcome. I do not understand this sentence. What does problematic mean?
>
> The term "problematic" is used in the sense that it leads to discrimination. For example, in hiring for jobs where physical fitness scores are important, if the decision is based solely on physical fitness scores, it would be unequal to exclude gender causation, as physical fitness is heavily influenced by gender differences between men and women. It is necessary to take into account that biologically different men and women have different fitness scores. Therefore, the path from gender to physical strength should be fair and other pathways is unfair.
>
>
> > I do not understand why $Y_{A \Leftarrow 0}$ does not use unfair pathways. Why is it not $Y_{A \Leftarrow 0 \| \pi}$?
>
> In Path-specific counterfactual fairness definition, the value of $A$ on an original data is assumed to be 0. As $Y_{A \Leftarrow 0}$ is the prediction of the original data, we do not care about unfair pathway $\pi$.
> To ensure that there is no influence from unfair paths when changing A from 0 to 1, it is necessary to consider $Y_{A \Leftarrow 1 \| \pi} - Y_{A \Leftarrow 0}$.
> On the other hand, when $A$ of an original data is 1, it is necessary to compute $Y_{A \Leftarrow 0 \| \pi} - Y_{A \Leftarrow 1}$.
>
>
> > the publisher must publicly disclose the causal graph $G$
>  that underpins the predictive model $f$. I am not very familiar with counterfactual fairness. However, it seems to me that most publishers simply fit their model to data without imposing a causal structure. That means they will not be able to provide such a causal graph. In this case, it seems methods such as the proposed one cannot be applied (?).
>
> Since this method is based on a causal graph, this method will not work well if a causal graph is not submitted from publisher.
>
>  > I do not understand why the EM algorithm is required to infer the noise variables if we assume additive noise. Specifically, $e_i = x_i - f_i(PA(x_i))$. Or are the latent variables for the EM algorithm not the noise variables, but some model parameters?  If yes, it would be interesting to know what latent variable model was fit here.
>
> Yes, we use the EM algorithm to address potential misspecification in models of mediators by capturing the exogenous variables as the latent variables. we will describe what the latant variable model was fit.
>
>  >  Do these error bars correspond to 1 std.? If yes, please add that for clarity. Otherwise, please write what exactly the bars correspond to.
>
> Yes. the error bars correspond to 1 std. We clarify this in our manuscript.

---

> > ### Author Response · Authors · 2025-01-30
> >
> > We sincerely appreciate your detailed and insightful comments.
> > We revised our manuscript based on your feedback, and the changes are highlighted in red.
> >
> > **Introduction**
> > > _Their framework assumes that any causal effect of the sensitive attribute on the outcome as inherently problematic and aims to eliminate any influence that the sensitive attribute might have on the outcome._ I do not understand this sentence. What does problematic mean? If we assume that no attributes should be causal for the prediction (we eliminate the influences), we end up without any attributes at all. Disclaimer: I am not very familiar with the framework by [1].
> >
> > We added an explanation why it is problematic in Section 1.
> >
> > > _we introduce the Möbius inversion formula_
> > It does not seem like the authors introduce the formula. It seems instead that they introduce this formula to the field of counterfactual fairness. Please clarify.
> >
> > we clarified this in the introduction section.
> >
> > **Section 3.1**
> > > _Since we focus on the impact of the sensitive attribute, we only consider exogenous variables that are not influenced by the sensitive attribute._ I am a bit confused by this sentence. From my understanding, exogenous variables are never influenced by endogenous variables. Please clarify.
> >
> > We removed the sentence because it was unnecessary.
> >
> >
> > **Section 3.2**
> > > In the definition of total effect of alternation (TE), I found it hard to see what the difference ...
> >
> > We provided additional explanation about path-specific counterfactual fairness, which is detailed above Equation 1.
> >
> > **Section 4**
> > > I do not understand why $Y_{A \Leftarrow 0}$ does not use unfair pathways. Why is it not
> > $Y_{A \Leftarrow 0 \| \pi}$? The unfair mechanisms are also used here.
> >
> > We added the explanation about it before equation 1.
> >
> > > _This is important because, when the publisher can define their causal models, the values of mediators can be adjusted as desired when altering the sensitive attribut_ I do not understand this sentence. Unfortunately, it seems relevant, because it seems to be to motivation for the proposed method. Please add more explanation here, maybe using the concrete example from section 3.1.
> >
> > For instance, if the publisher sets up a model where the sensitive attribute (gender) influences the mediator (income), they can adjust how much income changes when the sensitive attribute flips from male to female.
> > This adjustment can lead to a manipulated fairness outcome, as the publisher has the ability to tweak the system to make it appear fairer or more biased based on how they model the interaction between gender and income.
> > This ability to influence the intermediate mediators could distort the auditor's ability to evaluate fairness accurately.
> > We added this in Section 4.
> >
> > **Section 5.1**
> > > Not a requested change, but I have a feeling that this interior operator could be related to backtracking counterfactuals ([2, 3]),
> >
> > Thank you for providing the references. The concept of backtracking counterfactuals seems be related to the concept of counterfactual fairness rather than interior operator.
> >
> > **Section 5.2**
> > > _Intuitively,_ $\Delta_{v_{\mathcal{F}}}$ _indicates that the synergy effect of combining with T features_ This sentence makes no sense (wrong grammar and no content) and I believe it would make sense to spend much more text on providing the reader with an intuition for this variable. I find this variable very hard to grasp just like that.
> >
> > We have rewritten the sentence after equation 6.
> >
> > **Section 6**
> > > German datasets. The reference for the data set is missing.
> >
> > We added the URL of the dataset.
> >
> > > I do not understand why the EM algorithm is required to infer the noise variables if we assume additive noise. Specifically, $e_i =x_i - f_i(PA(x_i))$. Or are the latent variables for the EM algorithm not the noise variables, but some model parameters? If yes, it would be interesting to know what latent variable model was fit here.
> >
> > In our research, the noise variable is treated as a latent variable. The EM algorithm is used to infer these latent noise variables, as they are not directly observable. While the noise is assumed to be additive, its specific values must still be estimated from the data.
> > We added this in Section
> >
> >
> > > Do these error bars correspond to 1 std.? If yes, please add that for clarity.
> >
> > We added a description that error bars(regions) corresponds to 1 std in Section 6.1.

---

> > > ### Comment · Reviewer_Ntd9 · 2025-02-03
> > >
> > > I thank the authors for their response and the revisions made to the manuscript. I am satisfied with most of the adjustments. While I still find the discussion of the Möbius inversion formula somewhat meager, the revised sentence is at least clear now. Overall, I now recommend acceptance.

---

### Review · Reviewer_rw6V · 2024-12-19

**Summary Of Contributions:**

The paper focuses on the problem of path-specific counterfactual fairness in machine assisted decision support. In this context, the authors assume an underlying causal graph of the attributes and mediators that affect the final decision (e.g. to hire an individual), where some pathways (paths from attributes and mediators to the final decision) may be considered fair or unfair. Under this setup, the authors propose a method to adjust the output of a model predicting the final decision in order to achieve path-specific counterfactual fairness, i.e., to nullify the causal effect due to unfair pathways. To this end, the authors build upon techniques in the field of cooperative games and propose a post-training method to recompute the output of the model while removing the causal effect due to unfair mediators. For their method they propose a set function that captures parent relations between the  variables in the causal graph, which they use to compute the causal effects due to unfair pathways by decomposing it as a sum of Harsanyi dividends.

**Audience:**

Yes

**Broader Impact Concerns:**

The work already includes a broader impact statement the sufficiently addresses concerns and ethical implications.

**Claims And Evidence:**

Yes

**Requested Changes:**

Although the presentation issues do not generally reduce the importance of the contribution, the following changes appear necessary to recommend acceptance:
i) A clear presentation of experimental results in Figure 5, by adding, for example, a legend indicating the compared methods

ii) Correcting the definition of potential outcomes in section 3.1

Other changes that could improve the work include:

i) Improving the flow in the contribution bullets in the introduction, by first describing the proposed method and then perhaps highlighting its efficiency. The authors may even like to remove the bullets completely and have one coherent paragraph instead.

ii) Clarifying the meaning of endogenous variables in section 3.1.

iii) The authors discuss the selection of the reference value $r$ in case of a binary sensitive attribute. It may be worth discussing also how should one select the reference value in case of a sensitive attribute that can take more than two values from a discrete set.

iv) Fixing all typos
    - Abstract: ‘We empirically show that proposed algorithm” —> ‘We empirically show that our proposed algorithm”

    - Introduction, paragraph before bullet points “that effectively capture” —> “that effectively captures”

    - Introduction, paragraph before bullet points: “1dividends” —> “dividends”

    - Related work in the end “in the analyze” —> “in the analysis”

    - Paragraph above preliminary “it also have” —> “it also has”

    - Section 5.1, 1st paragraph “requires for establishing” —> “is required for”

    - Below Example 2 “a cause occur” —> “a cause occurs”

    - Pg 7 last line, “effect of combining with T features.”—> it is unclear with what the $T$ features are combined (perhaps “with” is not necessary).

    - Bottom of page 7, “$en(T)$ is a set of nodes that do not have no child in $T$” —> Perhaps the authors meant “$en(T)$ is a set of nodes that does not have any child in $T$”

    - Section 5.4, 3rd line “eahc” —> “each”

    - “A notable result here is that the our post-process approach” —> “A notable result here is that our post-process approach”

    - Figures 6 and 7 could have been aligned

    - Section 6.3 “the other base model” —> “the other base models”

    - Conclusion “we propose causally aware set function” —> “we propose a causally aware set function”

**Strengths And Weaknesses:**

**Strengths.** The proposed approach appears quite interesting and novel, as the work seems to be the first to build upon techniques on cooperative games to achieve path-specific counterfactual fairness. The authors sufficiently discuss related work and provide several examples to provide the intuition for their method. The technical results are mostly clearly presented  and the experimental results seem to include evaluation with both synthetic and real data.

**Weaknesses.** There seems to be room for improvement in terms of clarity and presentation. Some parts of the work seem as if written in a rush as there are quite a few parts that could be more clearly presented and there are several typos throughout the manuscript. In particular, the bulleted list in the introduction seems to lack flow as the first bullet highlight the efficiency of the proposed method while the second bullet actually described the method. In Section 3.1, in the description of the endogenous variables, it is unclear to what ‘quantification’ and ‘faculty type’ refer to.  In the same section, there also seems to be a typo in the definition of potential outcomes in the left term where $f_{M}(0,..)$ looks as if it should have been $f_{M}(1,..)$. In experiments, Figure 5 seems to lack a legend, therefore the reader can only speculate to which method  each line corresponds. In addition, the authors state that ‘The error bars representing the standard deviation’. However, the figures do not show error bars, but rather colored regions.

---

> ### Author Response · Authors · 2025-01-15
> **Response to Reviewer rw6V**
>
> Thank you for your insightful comments. Based on your suggestions, we will correct all typos, missing legends and error bars that you commented.
> > In Section 3.1, in the description of the endogenous variables, it is unclear to what ‘quantification’ and ‘faculty type’ refer to.
>
>    Thank you for pointint this out. It is a mistake to say “qualification” instead of “quantification.
>    Faculty type indicates the faculty to which the applicant belonged.
>    We will specify these in our manuscript.
>
>  > In the same section, there also seems to be a typo in the definition of potential outcomes in the left term where $f_M(0,...)$ looks as if it should have been $f_M(1,....)$.
>
>  The definition is correct because the value of the mediator $M$ does not affect the change of $A$, as it is not on any unfair pathway.
>  Then the value of $M$ keeps 0 in $Y_{A \Leftarrow 1\| \pi}$, which is the original value of a given data point.

---

> > ### Author Response · Authors · 2025-01-30
> >
> > We sincerely appreciate your detailed and insightful comments.
> > We revised our manuscript based on your feedback, and the changes are highlighted in red.
> >
> > > i) A clear presentation of experimental results in Figure 5, by adding, for example, a legend indicating the compared methods
> >
> > We added a legend in Figure 5.
> >
> > > ii) Correcting the definition of potential outcomes in section 3.1.
> >
> > We described more detailed explanation about the definition of potential outcomes in Section 3.1.
> >
> > > i) Improving the flow in the contribution bullets in the introduction, by first describing the proposed method and then perhaps highlighting its efficiency. The authors may even like to remove the bullets completely and have one coherent paragraph instead.
> >
> > We modified the contribution bullets as you commented in the end of Section 1.
> >
> > > ii) Clarifying the meaning of endogenous variables in section 3.1.
> >
> > we added an explanation about the endogenous variables in section 3.1.
> >
> > > iii) The authors discuss the selection of the reference value in case of a binary sensitive attribute. It may be worth discussing also how should one select the reference value in case of a sensitive attribute that can take more than two values from a discrete set.
> >
> > We have focused on changing the gender attribute from 0 to 1 in Section 5.1.
> > However, this approach is not limited to binary attributes; it also applies to sensitive attributes that can take more than two values from a discrete set.
> > Nonetheless, when the sensitive attribute has more than two values, it is important to define the fairness criteria carefully, as they may not be uniquely determined in such cases.
> > We added this in the end of Section 5.1.

---

### Review · Reviewer_ipuU · 2024-12-31

**Summary Of Contributions:**

The paper proposes a post-processing approach for achieving path-specific counterfactual fairness for an already developed predictive model and the availability of the causal graph used for developing the predictive model. This is achieved using tools from number theory to obtain dividends where each dividend corresponds to a causal effect along a certain pathway in the causal graph. The effect along unfair pathways is removed by removing the corresponding dividend.

**Audience:**

Yes

**Broader Impact Concerns:**

Since this is a post processing approach, it would be beneficial to address how to test whether the unfair paths are accurate and if they are susceptible to the location where the model is deployed.  Including a section that highlights the limitations of assuming that unfair pathways remain consistent across locations and are inherently accurate could add valuable context.

**Claims And Evidence:**

Yes

**Requested Changes:**

I have some questions and suggestions listed here.
1. Does the X-axis in Figure 5 correspond to $\epsilon$? If so, what is the reason for the range (0-0.2)?
2. Add a legend to highlight the model represented by each color.
3. It is unclear why the counterfactual effect (CE) was used to compare the different post-processing approaches. Removing the entire effect of the sensitive attribute may not be desirable.
4. Figure 6 is hard to interpret, considering that the X-axis scale for the proposed approach is different from the CF methods, making it hard to understand the chosen range. Ideally, if the counterfactual effect is based only on the causal graph, then the X-axis should be the same for all the methods. I’d appreciate some clarification on this.
5. Algorithm 1 is difficult to parse by itself; adding what $\epsilon$ and $v(P_{unfair})$ means there would improve the readability.
6. There’s a typo in dividends just before the contributions are listed in the introduction.
7. In the introduction, please describe what the dividends represent.
8. Add that the potential outcome for A set to 0 doesn’t require dependence on the unfair paths $pi$ for better readability before equation 1.
9. Instead of saying the sensitive attribute is flipped from 0 to 1, consider stating that the sensitive attribute is intervened on to set to a particular value.
10. In section 5, it isn't easy to parse what feasible sets, Mobius inversion formula, and dividends mean initially; I suggest improving the writing here to help the reader.
11. I found understanding how the reference value is selected difficult. Consider adding some explanation for this.
12. In example 1, does N represent {A,Q,D,M}?
13. It’s not entirely clear what the synergy effect is at the end of page 7.

**Strengths And Weaknesses:**

Strengths:

1. The problem studied is important, especially when auditing predictive models for fairness.
2. The idea of dividend correction is novel and has not been previously studied in the context of path-specific counterfactual fairness.
3. Experiments on synthetic as well as multiple real-world datasets help to assess the efficacy of the proposed approach.

Weakness:

1. The writing could be improved for better readability and to make the paper accessible to the audience not familiar with certain concepts introduced in the paper.
2. Certain experimental details are not adequately discussed, such as the choice of $\epsilon$; I have questions about this listed below.
3. It would be helpful to discuss the approach's limitations based on the number of nodes in the causal graph.

---

> ### Author Response · Authors · 2025-01-15
> **Responce to Reviewer ipuU**
>
> Thank you for your insightful comments. We will revise our manuscripts based on your suggestions.
>
> > Comment 1. Does the X-axis in Figure 5 correspond to \epsilon? If so, what is the reason for the range (0-0.2)?
>
> The x-axis does not mean $\epsilon$, but it shows PSE, which is obtained by evaluating the $Y_{A \Leftarrow 1 \| \pi} - Y_{A \Leftarrow 0}$.
>
> > Comment 3. It is unclear why the counterfactual effect (CE) was used to compare the different post-processing approaches.
>
> This is because the approach proposed in [Wu et al., 2019a] is the only post-process approach for causal fairness settings.
>
> > Comment 4. Figure 6 is hard to interpret
>
> As we wrote in our answer in Q1, the x-axis is the PSE, the result obtained, not the value of the parameter, so the x-axis compared to the previous experiment is different. We will explain more about the x-axis.
>
> > Comment 12. In example 1, does N represent {A,Q,D,M}?
>
> Yes, $N = \\{A,Q,D,M \\}$. We will reveal this in our manuscript.

---

> > ### Author Response · Authors · 2025-01-30
> >
> > We sincerely appreciate your detailed and insightful comments.
> > We revised our manuscript based on your feedback, and the changes are highlighted in red.
> >
> > > Add a legend to highlight the model represented by each color.
> >
> > We added a legend to Figure 5.
> >
> > > It is unclear why the counterfactual effect (CE) was used to compare the different post-processing approaches.Removing the entire effect of the sensitive attribute may not be desirable.
> >
> > We added a reason for this to the beginning of Section 6.2.
> >
> > >5. Algorithm 1 is difficult to parse by itself; adding what $\epsilon$ and $P_{\text{unfair}}$ means there would improve the readability.
> >
> > We added an explanation of $P_{\text{unfair}}$ and $\epsilon$ to Section 5.4 and Algorithm 1.
> >
> > > 7. In the introduction, please describe what the dividends represent.
> > The dividend can be intuitively explained as additional value created by changing specific attributes, after accounting for the overlapping contributions of its smaller subgroups of the attributes.
> >
> > The dividend can be intuitively understood as additional value created by modifying specific attributes, after considering the overlapping contributions of their smaller attribute subsets.
> > We added this in the introduction.
> >
> >
> > > 8. Add that the potential outcome for A set to 0 doesn’t require dependence on the unfair paths for better readability before equation 1.
> >
> > We added explanation about it before equation 1.
> >
> > > 9. Instead of saying the sensitive attribute is flipped from 0 to 1, consider stating that the sensitive attribute is intervened on to set to a particular value.
> >
> > Since the PSCF is designed for binary sensitive attributes, it needs to be generalized to accommodate non-binary sensitive attributes.
> > We described about this after equation 1.
> >
> > > 10. In section 5, it isn't easy to parse what feasible sets, Mobius inversion formula, and dividends mean initially; I suggest improving the writing here to help the reader.
> >
> > The key components of our algorithm are feasible sets, dividends, and M\"obius inversion formula.
> > The feasible sets represent subsets that capture causal relationships, and the dividends quantify  additional contribution by modifying specific attributes, accounting for the overlapping contributions of their smaller attribute subsets.
> > The M\"obius inversion formula serves as the method for computing these dividends.
> > We added this in the begging of Section 5.
> >
> >
> > >11. I found understanding how the reference value is selected difficult. Consider adding some explanation for this.
> >
> > The reference value accounts for changes in other attributes when the sensitive attribute is modified. It reflects how mediators are influenced by the sensitive attribute, ensuring an accurate measurement of causal effects.
> > Since we do not have the true causal model, we need to learn the mediators' models $f_D,f_M$ to accurately estimate the reference values.
> >
> > >12. In example 1, does N represent {A,Q,D,M}?
> >
> > We replaced $N$ to ${A,Q,D,M}$ in Example 1.
> >
> > >13. It’s not entirely clear what the synergy effect is at the end of page 7.
> >
> > The synergy effect corresponds to a pure contribution created by changing specific attributes, after accounting for the overlapping contributions of its smaller subgroups of the attributes. We described this after equation 6.

---

> > > ### Comment · Reviewer_ipuU · 2025-02-03
> > >
> > > I thank the authors for their detailed response. I am satisfied with most of the adjustments. The clarifications improve the overall readability and contribution of the paper. I recommend acceptance.

---

### Decision · Action_Editor_JKSx · 2025-02-03

**Recommendation:** Accept as is

**Comment:**

The core ideas in the submission as a novel approach path-specific fairness were perceived as interesting and strong. Throughout the discussion, the authors substantially improved the manuscript such that most questions and request from the reviewers were addressed. Overall, the submission is now in a solid state for publication at TMLR.

**Audience:**

Yes, this paper clearly addresses (parts of) the TMLR audience.

**Claims And Evidence:**

In particular after the discussion, the claims made in the submission are supported by convincing and clear evidence.